

# Single-boson exchange formulation of the Schwinger-Dyson equation and its application to the functional renormalization group

Miriam Patricolo[1,2,3⋆], Marcel Gievers[4,5], Kilian Fraboulet[1,6],
Aiman Al-Eryani[7], Sarah Heinzelmann[6], Pietro M. Bonetti[3,8],
Alessandro Toschi[2], Demetrio Vilardi[3] and Sabine Andergassen[1,2]

**1** Institute of Information Systems Engineering,
Vienna University of Technology, Vienna, Austria
**2** Institute for Solid State Physics, Vienna University of Technology, Vienna, Austria
**3** Max Planck Institute for Solid State Research, Heisenbergstrasse 1, Stuttgart, Germany
**4** Arnold Sommerfeld Center for Theoretical Physics, Center for NanoScience,
and Munich Center for Quantum Science and Technology,
Ludwig-Maximilians-Universität München, München, Germany
**5** Max Planck Institute of Quantum Optics, Garching, Germany
**6** Institute for Theoretical Physics and Center for Quantum Science,
Universität Tübingen, Tübingen, Germany
**7** Theoretical Physics III, Ruhr-University Bochum, 44801 Bochum, Germany
**8** Department of Physics, Harvard University, Cambridge MA 02138, USA

⋆ miriam.patricolo@tuwien.ac.at

## Abstract

We extend the recently introduced single-boson exchange formulation to the computation of the self-energy from the Schwinger–Dyson equation (SDE). In particular, we derive its expression both in diagrammatic and in physical channels. The simple form of the single-boson exchange SDE, involving only the bosonic propagator and the fermion-boson vertex, but not the rest function, allows for an efficient numerical implementation. We furthermore discuss its implications in a truncated unity solver, where a restricted number of form factors introduces an information loss in the projection of the momentum dependence that in general affects the equivalence between the different channel representations. In the application to the functional renormalization group, we find that the convergence in the number of form factors depends on the channel representation of the SDE. For the two-dimensional Hubbard model at weak coupling, the pseudogap opening driven by antiferromagnetic fluctuations is captured already by a single ($s$-wave) form factor in the magnetic channel representation, differently to the density and superconducting channels.

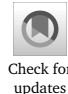

# 1 Introduction

The recently introduced single-boson exchange decomposition [1] provides a valuable tool in the quantum field-theoretic treatment of quantum many-body systems [2–12]. It features a physically intuitive and also computationally efficient description of the relevant fluctuations in terms of processes involving the exchange of a single boson, describing a collective excitation, and a residual part containing the multiboson processes. The effective bosonic interaction is represented by bosonic propagators and fermion-boson couplings also referred to as Yukawa couplings or Hedin vertices [13] determined from the vertex asymptotics, in analogy to the construction of the kernel functions defining the high-frequency asymptotics [14].

At weak coupling, this effective bosonic interaction yields quantitatively accurate results, while the multiboson contributions are irrelevant and can be neglected [15], allowing for a substantial reduction of the computational complexity of the vertex function: Since the multi-boson processes are the only ones to depend on three independent momentum and frequency variables, neglecting them drastically reduces the computational complexity of the problem. In contrast, the bosonic propagators and fermion-boson couplings depend on one and two independent arguments, respectively, and therefore their numerical treatment including the full momentum and frequency dependence is much less demanding.

At strong coupling, the advantages of the single-boson exchange formalism are particularly prominent in the non-perturbative regime of intermediate to strong electron-electron interaction. In fact, these interaction values lead to multiple divergences in the two-particle irreducible vertex functions [16–30], which makes the applicability of conventional Bethe–Salpeter equations and/or parquet formalism [31,32] beyond the weak-coupling regime rather problematic. In the single-boson exchange formulation of the diagrammatics, instead, the corresponding irreducible vertex functions are defined in a different way: They are obtained from

the difference between the full vertex and the single-boson exchange diagrams, each of which is composed of diagrams that correspond to physical correlators up to an amputation of the external legs. Beyond providing a much more transparent link to the underlying physics than the parquet formalism, no diagrammatic element of the single-boson exchange decompositions of the vertex function displays [1, 2] the non-perturbative divergencies which plague their parquet counterparts.

We here provide a unified framework for the consistent derivation of the Schwinger–Dyson equation (SDE) for the self-energy in the single-boson exchange formulation. Its simpler form involves only the bosonic propagator and the fermion-boson vertex in a single channel and not the rest function. Moreover, the expression for the SDE derived within the single-boson exchange formalism has a one-loop structure, making its evaluation easier than the standard textbook expression. Notably, the possibility of using different but equivalent self-energy formulations in the various channels does not depend on a specific choice of the Fierz decoupling parameter, which is related to the Fierz ambiguity [33]. Moreover, the change of representation of the Schwinger–Dyson equation in the resulting triangular form is particularly useful for the postprocessing tool of the fluctuation diagnostic, which enables the quantification of the different fluctuation contributions. Specifically, this approach avoids the need for partial summations required in earlier methods [34–38]. On a more practical perspective, we also discuss the relevant implications for truncated unity (TU) solvers [4, 39–44], where the information loss in the form-factor projection of the momentum dependence generally affects the equivalence between the different channel representations. Specifically, we apply the single-boson exchange expression for the SDE to the functional renormalization group (fRG) [45, 46] and demonstrate that the self-energy flow determined by its derivative [47] captures the pseudogap opening in the two-dimensional (2D) Hubbard model at weak coupling. However, the different channel representations of the SDE converge differently in the number of form factors. The antiferromagnetic fluctuations dominating at half filling are best described in the magnetic channel in which the onset of the pseudogap opening is captured by using only the $s$-wave form factor.

The paper is structured as follows: we first introduce the formalism in Section 2. Specifically, the presented matrix representation of the spin structure allows for a compact notation to efficiently sum over the involved variables and indices, the technical details are reported in Appendix A. After a brief review of the single-boson exchange representation, we derive the form of the SDE as the main result of the present work. In Section 3, we showcase the application to the fRG. We present results for the 2D Hubbard model at weak coupling and discuss the implications arising in the implementation with TU solvers. Finally, we provide a summary of our findings and conclusions in Section 4.

## 2 Single-boson exchange formulation of the SDE

### 2.1 Conventional SDE and matrix formalism

Before reviewing the single-boson exchange representation, we present the formalism [9] applicable to any lattice fermion system with the classical action of the form

$$S[\overline{c}, c] = -\overline{c}_{1'} G_{0;1'|1}^{-1} c_1 - \frac{1}{4} U_{1'2'|12} \overline{c}_{1'} \overline{c}_{2'} c_2 c_1 \,. \tag{1}$$

The numbers $1', 2', 1, 2$ labelling the Grassmann fields $c_i$ represent generic indices, which enclose spin components, momenta, and Matsubara frequencies. For these, we use Einstein's convention, i.e., repeated indices are summed over. Furthermore, $G_0$ denotes the bare propagator and $U$ the crossing-symmetric [31, 32] bare interaction vertex $U_{1'2'|12} = -U_{2'1'|12} = -U_{1'2'|21}$.

We assume energy conservation and translational invariance resulting in momentum and frequency conservation.

The conventional form of the SDE for the self-energy is the main subject of the present work and reads [31, 32]

$$\Sigma_{1'|1} = -U_{1'2'|12}G_{2|2'} - \frac{1}{2}U_{1'3'|42}G_{2|2'}G_{3|3'}G_{4|4'}V_{4'2'|13}. \tag{2}$$

Here, $V$ is the full four-point interaction vertex. Equation (2) represents the starting point for the derivation of its single-boson exchange formulation, as presented in the next sections. The products of the Green's functions define the bubbles in a given channel

$$\Pi_{ph;12|34} = -G_{2|3}G_{1|4}, \qquad \Pi_{\overline{ph};12|34} = G_{1|3}G_{2|4}, \qquad \Pi_{pp;12|34} = \frac{1}{2}G_{1|3}G_{2|4}. \tag{3}$$

With these definitions, Eq. (2) can be rewritten as

$$
\begin{aligned}
\Sigma_{1'|1} &= -U_{1'2'|12}G_{2|2'} + \frac{1}{2}G_{4|4'}U_{1'3'|42}\Pi_{ph;32|2'3'}V_{4'2'|13} \\
&= -U_{1'2'|12}G_{2|2'} + \frac{1}{2}G_{4|4'}U_{3'1'|24}\left[\Pi_{ph}\circ V\right]_{4'2|13'} \\
&= U_{2'1'|12}G_{2|2'} + \frac{1}{2}G_{4|4'}\left[U\circ\Pi_{ph}\circ V\right]_{4'1'|14},
\end{aligned}
\tag{4}
$$

in the $ph$ channel. Omitting the indices, yields the compact form

$$\Sigma = G \cdot \left( U + \frac{1}{2}\left[U\circ\Pi_{ph}\circ V\right] \right), \tag{5}$$

where we introduced the $\circ$ product indicating the summation over spin indices, momenta, and frequencies [9, 48]. The channel-dependent product of four-point functions $A$ and $B$ is defined by

$$
\begin{aligned}
ph &: [A\circ B]_{12|34} = A_{62|54}B_{15|36}, & \text{(6a)} \\
\overline{ph} &: [A\circ B]_{12|34} = A_{16|54}B_{52|36}, & \text{(6b)} \\
pp &: [A\circ B]_{12|34} = A_{12|56}B_{56|34}. & \text{(6c)}
\end{aligned}
$$

Note that the product can be represented by matrices, see Appendix A for details. We furthermore used the product involving a (two-point) Green's function $G$ defined by

$$[A\cdot G]_{1'|1} = A_{1'2'|12}G_{2|2'} = -G_{2|2'}A_{2'1'|12} = -[G\cdot A]_{1'|1}. \tag{7}$$

For the definition of the loop product $\cdot$, the order of $G$ and $A$ is decisive since we absorb a minus sign originating from the crossing symmetry of the vertex $A$. Analogously, we can rewrite the second term on the right-hand side of Eq. (2) in the other diagrammatic channels. We obtain

$$
\begin{aligned}
U_{1'3'|42}G_{2|2'}G_{3|3'}G_{4|4'}V_{4'2'|13} &= G_{2|2'}U_{1'3'|42}\Pi_{\overline{ph};34|3'4'}V_{4'2'|13} \\
&= G_{2|2'}\left[U\circ\Pi_{\overline{ph}}\circ V\right]_{1'2'|12},
\end{aligned}
\tag{8}
$$

for the $\overline{ph}$ channel and

$$
\begin{aligned}
U_{1'3'|42}G_{2|2'}G_{3|3'}G_{4|4'}V_{4'2'|13} &= U_{1'3'|42}G_{3|3'}2\Pi_{pp;24|2'4'}V_{4'2'|13} \\
&= G_{3|3'}\left[U\circ 2\Pi_{pp}\circ V\right]_{1'3'|13},
\end{aligned}
\tag{9}
$$

Figure 1: Diagrammatic representation of the SDE for the self-energy: We show the diagram in the conventional form and the corresponding respresentation in single-boson exchange formalism in the $\overline{ph}$ and $pp$ channel (without the Hartree term).

for the $pp$ channel. Thus, Eq. (5) can be expressed equivalently as

$$\Sigma = -\left(U + \frac{1}{2}[U \circ \Pi_{\overline{ph}} \circ V]\right) \cdot G \tag{10a}$$

$$= -\left(U + [U \circ \Pi_{pp} \circ V]\right) \cdot G, \tag{10b}$$

see Fig. 1 for their diagrammatic representation. We note that the sign change in the Hartree term is due to the reverted order of the product, see also Eq. (7). Equations (5) and (10) are the starting point for the derivation of the SDE in the single-boson exchange representation.

## 2.2 Single-boson exchange representation

The single-boson exchange decomposition of the two-particle vertex is based on an alternative notion of reducibility, known as $U$ reducibility, where $U$ is the bare interaction [1]. The concept builds on the observation of the primary bosonic dependence of diagrams and their interpretation as exchange of a single boson. Diagrams falling into this category are termed $U$-reducible as they can be divided into two parts by cutting a bare interaction. Conversely, diagrams that cannot be divided this way are termed $U$ irreducible. Similarly to the two-particle reducibility underlying the classification of diagrams in the parquet formalism [31,32], the $U$-reducible diagrams can be further categorized on whether the two lines connected to the bare interaction are particle-particle ($pp$), particle-hole ($ph$), or particle-hole crossed ($\overline{ph}$) lines. Note that a $U$-reducible diagram is also two-particle reducible, with the exception of the bare interaction itself, which is considered $U$-reducible in all three channels.

Exploiting momentum and frequency conservation for one-particle correlators, such as the Green's function, gives

$$G_{\sigma_{1'}|\sigma_1}(k_{1'}|k_1) = \delta_{\sigma_{1'}|\sigma_1}\delta_{\mathbf{k}_{1'},\mathbf{k}_1}\delta_{\nu_{1'},\nu_1}G_{\sigma_{1'},\sigma_1}(k_1). \tag{11}$$

For two-particle objects, such as the full two-particle vertex, we have

$$V_{\sigma_{1'}\sigma_{2'}|\sigma_1\sigma_2}(k_{1'},k_{2'}|k_1,k_2) = \delta_{\mathbf{k}_{1'}+\mathbf{k}_{2'},\mathbf{k}_1+\mathbf{k}_2}\delta_{\nu_{1'}+\nu_{2'},\nu_1+\nu_2}V_{\sigma_{1'}\sigma_{2'}|\sigma_1\sigma_2}(Q_r,k_r,k_r'), \tag{12}$$

where the channel $r$ defines the bosonic $Q_r = (\mathbf{Q}_r, \Omega_r)$ and fermionic arguments $k_r = (\mathbf{k}_r, \nu_r)$ and $k_r' = (\mathbf{k}_r', \nu_r')$, see also Fig. 7 in Appendix A for the definitions of $k_r$ and $Q_r$ in the respective channels $r$.

Specifically, the latter applies also for the bare interaction vertex $U_{\sigma_{1'}\sigma_{2'}|\sigma_1\sigma_2}(k_{1'},k_{2'}|k_1,k_2)$. Through Eqs. (11)–(12), one-particle objects only depend on one momentum and frequency variable, while two-particle objects in general depend on three.

The sum of all $U$-reducible diagrams in a given channel $r = pp, ph, \overline{ph}$ including the bare interaction is given by

$$\nabla_r = \bar{\lambda}_r \bullet w_r \bullet \lambda_r, \tag{13}$$

where the $\bullet$ product indicates the summation over spin indices only (with the same definition as in Eqs. (6), but excluding the summation over momenta and frequencies). It represents the

exchange of a single bosonic propagator $w_r$ between two fermion-boson couplings $\lambda_r$ and $\bar{\lambda}_r$. Diagrams that are two-particle reducible, but $U$ irreducible with respect to the channel $r$ do not fall into this category. They are collected in the rest function $M_r$ containing the multiboson exchange processes (see Fig. 1 in [10] and Fig. 5 in [9] as examples). In the notation introduced above, both $\lambda_r$ and $w_r$ are four-point objects with respect to the spin indices. For the reduced frequency and momentum dependence of the single-boson exchange vertices, it is essential that the bare interaction $U$ does not depend on frequencies and momenta. In particular, this is the case for an instantaneous local $U$. Explicitly, the bosonic propagator $w_r = w_r(Q_r)$ then depends on a single bosonic argument and $\lambda_r = \lambda_r(Q_r, k_r)$ on both a bosonic and a fermionic argument in the presence of momentum and frequency conservation.

In the following, we will exploit the relation [9]:

$$w_r \bullet \lambda_r = U + U \circ \Pi_r \circ V \tag{14}$$

($\bar{\lambda}_r \bullet w_r = U + V \circ \Pi_r \circ U$ respectively), which is crucial in the derivation of the SDE in single-boson exchange representation. This relation applies for local interactions, while the generalization to non-local interactions is briefly discussed in Appendix B.

## 2.3  Derivation in diagrammatic channels

We first determine the SDE for the self-energy in diagrammatic channels. In the single-boson exchange formulation, its form turns out to be particularly simple and hence more advantageous for the numerical implementation.

For the spin $\uparrow$ component (the $\downarrow$ component is obtained straightforwardly by inverting the spin indices), Eqs. (5) and (10) for the different channels read

$$\Sigma^\uparrow = G^\downarrow \cdot U^{\widehat{\downarrow\uparrow}} + \frac{1}{2} G^\uparrow \cdot \left[ U \circ \Pi_{ph} \circ V \right]^{\uparrow\uparrow} + \frac{1}{2} G^\downarrow \cdot \left[ U \circ \Pi_{ph} \circ V \right]^{\widehat{\downarrow\uparrow}}, \tag{15a}$$

$$\Sigma^\uparrow = -U^{\uparrow\downarrow} \cdot G^\downarrow - \frac{1}{2} [U \circ \Pi_{\overline{ph}} \circ V]^{\uparrow\uparrow} \cdot G^\uparrow - \frac{1}{2} [U \circ \Pi_{\overline{ph}} \circ V]^{\uparrow\downarrow} \cdot G^\downarrow, \tag{15b}$$

$$\Sigma^\uparrow = -U^{\uparrow\downarrow} \cdot G^\downarrow - [U \circ \Pi_{pp} \circ V]^{\uparrow\uparrow} \cdot G^\uparrow - [U \circ \Pi_{pp} \circ V]^{\uparrow\downarrow} \cdot G^\downarrow, \tag{15c}$$

where we used $U^{\uparrow\uparrow} = 0$ for local interactions and introduced the short-hand notation

$$\Sigma^\sigma = \Sigma^{\sigma|\sigma}, \qquad U^{\sigma\sigma} = U^{\sigma\sigma|\sigma\sigma}, \qquad U^{\sigma\overline{\sigma}} = U^{\sigma\overline{\sigma}|\sigma\overline{\sigma}}, \qquad U^{\widehat{\sigma\overline{\sigma}}} = U^{\sigma\overline{\sigma}|\overline{\sigma}\sigma}, \tag{16}$$

with $\sigma = \uparrow / \downarrow$ and $\overline{\uparrow} = \downarrow$, $\overline{\downarrow} = \uparrow$. We here assume only U(1) symmetry in order to account for a magnetic field. We will restrict ourselves to the SU(2) symmetric case for the derivation in physical channels in Sec. 2.4, where we exploit $\Sigma^\uparrow = \Sigma^\downarrow$, $G^\uparrow = G^\downarrow$, $V^{\uparrow\uparrow} = V^{\uparrow\downarrow} + V^{\widehat{\uparrow\downarrow}}$ and $V^{\uparrow\downarrow} = V^{\downarrow\uparrow}$ [31,32]. In Eqs. (15), only the spin indices are reported explicitly, whereas the full momentum and frequency dependence is determined below. As a general rule, the sum in the products includes all indices except for the specified ones (in this case, the spin indices have already been summed over). In the following, we first focus on the $ph$ channel and then extend our results to the $\overline{ph}$ and $pp$ channels. The summation over the spin indices in Eq. (15a) yields

$$\Sigma^\uparrow = G^\downarrow \cdot U^{\widehat{\downarrow\uparrow}} + G^\downarrow \cdot U^{\widehat{\downarrow\uparrow}} \circ \Pi_{ph}^{\widehat{\downarrow\uparrow}} \circ V^{\widehat{\downarrow\uparrow}}, \tag{17}$$

where we used that $\Pi_{ph}^{\uparrow\downarrow} = 0$ due to the matrix structure and

$$G^\uparrow \cdot U^{\downarrow\uparrow} \circ \Pi_{ph}^{\downarrow\downarrow} \circ V^{\uparrow\downarrow} = G^\downarrow \cdot U^{\widehat{\downarrow\uparrow}} \circ \Pi_{ph}^{\widehat{\downarrow\uparrow}} \circ V^{\widehat{\downarrow\uparrow}}, \tag{18}$$

as a consequence of crossing symmetry. The latter is obtained by applying the relation (A.3) discussed in Appendix A to both the bare interaction U and the full vertex V.

We now express our findings in the single-boson exchange formalism. Using the relation outlined in Eq. (14), we can express the product $U^{\widehat{\downarrow\uparrow}} \circ \Pi_{ph}^{\widehat{\downarrow\uparrow}} \circ V^{\widehat{\downarrow\uparrow}}$ in Eq. (17) as

$$\left[ U \circ \Pi_{ph} \circ V \right]^{\widehat{\downarrow\uparrow}} = \left[ w_{ph} \bullet \lambda_{ph} - U \right]^{\widehat{\downarrow\uparrow}}. \tag{19}$$

Performing the spin summations yields

$$\Sigma^{\uparrow} = G^{\downarrow} \cdot (w_{ph}^{\widehat{\downarrow\uparrow}} \lambda_{ph}^{\widehat{\downarrow\uparrow}}), \tag{20}$$

for the self-energy in the $ph$ channel. We note that in contrast to Eq. (17), the Hartree term does not explicitly appear anymore, since it is absorbed in the translation to the single-boson exchange representation through $w_{ph}$ and $\lambda_{ph}$ by (19).

Analogous steps allow us to rewrite Eqs. (15b) and (15c) for the $\overline{ph}$ and the $pp$ channel, respectively

$$\Sigma^{\uparrow} = -(w_{\overline{ph}}^{\uparrow\downarrow} \lambda_{\overline{ph}}^{\uparrow\downarrow}) \cdot G^{\downarrow}, \tag{21a}$$

$$\Sigma^{\uparrow} = -[w_{pp}^{\uparrow\downarrow}(2\lambda_{pp}^{\uparrow\downarrow} - 1)] \cdot G^{\downarrow}, \tag{21b}$$

where we used the relations $w_{pp}^{\widehat{\uparrow\downarrow}} = -w_{pp}^{\uparrow\downarrow}$ and $\lambda_{pp}^{\widehat{\downarrow\uparrow}} = 1 - \lambda_{pp}^{\uparrow\downarrow}$.

We note that the SDE in single-boson exchange representation can also be obtained by directly applying Eqs. (14) to Eqs. (5) and (10), yielding

$$\Sigma = -(w_r \bullet \lambda_r) \cdot G. \tag{22}$$

The corresponding diagrammatic representations are shown in Fig. 1 for the $\overline{ph}$ and $pp$ channel representations (21). For the $pp$ channel, the product $w_{pp} \bullet \lambda_{pp}$ is determined by

$$\begin{bmatrix} [w_{pp} \bullet \lambda_{pp}]^{\uparrow\downarrow} & [w_{pp} \bullet \lambda_{pp}]^{\widehat{\uparrow\downarrow}} \\ [w_{pp} \bullet \lambda_{pp}]^{\widehat{\downarrow\uparrow}} & [w_{pp} \bullet \lambda_{pp}]^{\downarrow\uparrow} \end{bmatrix} = w_{pp}^{\uparrow\downarrow} \begin{bmatrix} 1 & -1 \\ -1 & 1 \end{bmatrix} \begin{bmatrix} \lambda_{pp}^{\uparrow\downarrow} & -(\lambda_{pp}^{\uparrow\downarrow} - 1) \\ -(\lambda_{pp}^{\uparrow\downarrow} - 1) & \lambda_{pp}^{\downarrow\uparrow} \end{bmatrix}$$

$$= w_{pp}^{\uparrow\downarrow} \begin{bmatrix} (2\lambda_{pp}^{\uparrow\downarrow} - 1) & -(2\lambda_{pp}^{\downarrow\uparrow} - 1) \\ -(2\lambda_{pp}^{\uparrow\downarrow} - 1) & (2\lambda_{pp}^{\downarrow\uparrow} - 1) \end{bmatrix}, \tag{23}$$

where the simple forms of the matrices result from crossing symmetry, see Appendix A. Specifying the spin component, the self-energy can be read off as

$$\Sigma^{\uparrow} = -[w_{pp}^{\uparrow\downarrow}(2\lambda_{pp}^{\uparrow\downarrow} - 1)] \cdot G^{\downarrow}. \tag{24}$$

However, for the $ph$ and $\overline{ph}$ channels the corresponding matrices have a more complex form and crossing symmetry can only be used at the level of Eqs. (15a) and (15b) to simplify the spin summations. We now provide the momentum and frequency dependence of the SDE for the self-energy in diagrammatic channels. Applying the momentum and frequency conventions, we determine the explicit forms of Eqs. (20) and (21) to be

$$\Sigma^{\uparrow}(\mathbf{k}; \nu) = \sum_{\mathbf{Q},\Omega} w_{ph}^{\widehat{\uparrow\downarrow}}(\mathbf{Q}; \Omega) \lambda_{ph}^{\widehat{\uparrow\downarrow}}\left(\mathbf{Q}, \mathbf{k} - \mathbf{Q}; \Omega, \nu - \left\lceil \frac{\Omega}{2} \right\rceil\right) G^{\downarrow}(\mathbf{k} - \mathbf{Q}; \nu - \Omega), \tag{25a}$$

$$\Sigma^{\uparrow}(\mathbf{k}; \nu) = -\sum_{\mathbf{Q},\Omega} w_{\overline{ph}}^{\uparrow\downarrow}(\mathbf{Q}; \Omega) \lambda_{\overline{ph}}^{\uparrow\downarrow}\left(\mathbf{Q}, \mathbf{k} - \mathbf{Q}; \Omega, \nu - \left\lceil \frac{\Omega}{2} \right\rceil\right) G^{\downarrow}(\mathbf{k} - \mathbf{Q}; \nu - \Omega), \tag{25b}$$

$$\Sigma^{\uparrow}(\mathbf{k}; \nu) = -\sum_{\mathbf{Q},\Omega} w_{pp}^{\uparrow\downarrow}(\mathbf{Q}; \Omega) \left[ 2\lambda_{pp}^{\uparrow\downarrow}\left(\mathbf{Q}, \mathbf{Q} - \mathbf{k}; \Omega, \left\lceil \frac{\Omega}{2} \right\rceil - \nu\right) - 1 \right] G^{\downarrow}(\mathbf{Q} - \mathbf{k}; \Omega - \nu), \tag{25c}$$

where the symbol $\lceil \cdots \rceil$ ($\lfloor \cdots \rfloor$) rounds its argument up (down) to the nearest bosonic Matsubara frequency. The corresponding equations for $\Sigma^\downarrow$ are obtained by reversing the spin indices. For the details on the derivation, we refer to Appendix C. Without any approximation, the three expressions of the SDE in single-boson exchange representation, Eqs. (25), are equivalent: the bosonic propagator and fermion-boson coupling from any single channel allows to reconstruct all self-energy diagrams. However, TU solvers expanding the fermionic momentum dependence in a finite number of form factors generally lead to different results for the various channels, as will be discussed below.

## 2.4 Derivation in physical channels

In this section, we translate the simple form of the SDE in single-boson exchange representation derived in diagrammatic channels to physical ones,[1] i.e., the magnetic, density, and superconducting channels, in which the single-boson exchange decomposition has been originally introduced [1]. These channels involve specific linear combinations of the spin components, designed to diagonalize the spin structure in the Bethe-Salpeter equations for systems with SU(2) symmetry [31, 32]. This offers interpretative advantages as it allows for a direct physical identification of the collective degrees of freedom at play.

Restricting ourselves to SU(2)-symmetric systems, in the shorthand notation introduced above, the six spin components of the full vertex reduce to $V^{\uparrow\downarrow}$, $V^{\widehat{\uparrow\downarrow}}$, $V^{\uparrow\uparrow}$, equivalent to $V^{\downarrow\uparrow}$, $V^{\widehat{\downarrow\uparrow}}$, and $V^{\downarrow\downarrow}$ respectively. Similarly, for the spin components of the self-energy and the Green's function holds $\Sigma^\uparrow = \Sigma^\downarrow$ and $G^\uparrow = G^\downarrow$. Furthermore, we have $V^{\uparrow\uparrow} = V^{\uparrow\downarrow} + V^{\widehat{\uparrow\downarrow}}$, as it follows from the definitions in Eqs. (16). We define the density, magnetic, and the superconducting channels as [49]

$$V^{\mathrm{M}} = V^{\uparrow\uparrow}_{ph} - V^{\uparrow\downarrow}_{ph} = -V^{\uparrow\downarrow}_{\widehat{ph}}, \tag{26a}$$

$$V^{\mathrm{D}} = V^{\uparrow\uparrow}_{ph} + V^{\uparrow\downarrow}_{ph} = 2V^{\uparrow\downarrow}_{ph} - V^{\uparrow\downarrow}_{\widehat{ph}}, \tag{26b}$$

$$V^{\mathrm{SC}} = V^{\uparrow\downarrow}_{pp}. \tag{26c}$$

The bosonic propagators $w$ in physical channels are determined by analogous relations. The same applies for the fermion-boson couplings $\lambda$ except for its expression in the superconducting channel, see below. Their inversion yields

$$w^{\widehat{\uparrow\downarrow}}_{ph} = w^{\mathrm{M}}, \qquad w^{\uparrow\uparrow}_{ph} = \frac{w^{\mathrm{M}} + w^{\mathrm{D}}}{2}, \qquad w^{\uparrow\downarrow}_{ph} = \frac{w^{\mathrm{D}} - w^{\mathrm{M}}}{2}, \qquad w^{\uparrow\downarrow}_{pp} = w^{\mathrm{SC}}, \tag{27a}$$

$$\lambda^{\widehat{\uparrow\downarrow}}_{ph} = \lambda^{\mathrm{M}}, \qquad \lambda^{\uparrow\uparrow}_{ph} = \frac{\lambda^{\mathrm{M}} + \lambda^{\mathrm{D}}}{2}, \qquad \lambda^{\uparrow\downarrow}_{ph} = \frac{\lambda^{\mathrm{D}} - \lambda^{\mathrm{M}}}{2}, \qquad \lambda^{\uparrow\downarrow}_{pp} = \frac{\lambda^{\mathrm{SC}} + 1}{2}, \tag{27b}$$

where we used the $\widehat{\uparrow\downarrow}$ component for the magnetic channel. We note that indeed $w^{\widehat{\uparrow\downarrow}}_r = w^{\uparrow\uparrow}_r - w^{\uparrow\downarrow}_r$. For the details on the superconducting fermion-boson coupling $\lambda^{\mathrm{SC}} = 2\lambda^{\uparrow\downarrow}_{pp} - 1$ differing from the corresponding one for the bosonic propagator, we refer to Appendix A. It is worth noting that the $pp$ channel allows to define both the singlet and triplet pairing channels

$$V^{\mathrm{s}} = V^{\uparrow\downarrow}_{pp} - V^{\widehat{\uparrow\downarrow}}_{pp}, \qquad V^{\mathrm{t}} = V^{\uparrow\downarrow}_{pp} + V^{\widehat{\uparrow\downarrow}}_{pp}. \tag{28}$$

Thus, the definition of the SC channel is consistent with

$$V^{\mathrm{SC}} = \frac{V^{\mathrm{s}} + V^{\mathrm{t}}}{2}. \tag{29}$$

---

[1]Ref. [9] illustrates the relationship between these "physical" and the "diagrammatic" channels assuming SU(2) spin symmetry.

Equations (28) hold also for the bosonic propagators $w^s$, $w^t$ and for the fermion-boson couplings $\lambda^s$, $\lambda^t$. The relation to the above expression for $\lambda^{SC}$ is obtained by considering $\lambda^{\uparrow\uparrow}_{pp} = 1$, see Appendix A. The singlet channel then reads

$$\lambda^s = \lambda^{\uparrow\downarrow}_{pp} - \lambda^{\widehat{\uparrow\downarrow}}_{pp} = \lambda^{\uparrow\downarrow}_{pp} - \lambda^{\uparrow\uparrow}_{pp} + \lambda^{\uparrow\downarrow}_{pp} = 2\lambda^{\uparrow\downarrow}_{pp} - 1\,, \tag{30}$$

which encodes the superconducting channel, while $\lambda^t = \lambda^{\uparrow\downarrow}_{pp} + \lambda^{\widehat{\uparrow\downarrow}}_{pp} = 1$.

Using the relations in Eqs. (27), both the $ph$ and $\overline{ph}$ formulations of the self-energy in Eqs. (20) and (21a) translate to

$$\Sigma = G \cdot (w^M \lambda^M)\,. \tag{31}$$

For the superconducting channel, Eq. (21b) yields

$$\Sigma = -(w^{SC} \lambda^{SC}) \cdot G\,. \tag{32}$$

In order to derive the density channel formulation, we have to start from the general form (15a). In the single-boson exchange formulation, it reads

$$\Sigma^\uparrow = \frac{1}{2}G^\downarrow \cdot U^{\widehat{\downarrow\uparrow}} + \frac{1}{2}G^\uparrow \cdot (w^{\uparrow\uparrow}_{ph}\lambda^{\uparrow\uparrow}_{ph} + w^{\downarrow\downarrow}_{ph}\lambda^{\uparrow\downarrow}_{ph}) + \frac{1}{2}G^\downarrow \cdot (w^{\widehat{\downarrow\uparrow}}_{ph}\lambda^{\widehat{\downarrow\uparrow}}_{ph})\,, \tag{33}$$

where we used Eq. (14). In presence of SU(2) symmetry, this translates to

$$\Sigma = \frac{3}{4}G \cdot (w^M \lambda^M) + \frac{1}{4}G \cdot (w^D \lambda^D) - \frac{1}{2}G \cdot U^D\,, \tag{34}$$

where we introduced $U^D = U^{\uparrow\downarrow}$ consistently with the density component of the bare vertex in Eqs. (26). The comparison with Eq. (31) then leads to

$$\Sigma = G \cdot (w^D \lambda^D) - 2G \cdot U^D\,. \tag{35}$$

This shows that the general form of Eq. (15a) is essential to derive the SDE in all three channels.

The explicit momentum and frequency dependence of the SDE in physical channels can be determined along the same lines as for the diagrammatic channels (for the details on the derivation see Appendix C) and reads

$$\Sigma(\mathbf{k}; \nu) = \sum_{\mathbf{Q},\Omega} w^M(\mathbf{Q};\Omega)\lambda^M\left(\mathbf{Q}, \mathbf{k}-\mathbf{Q}; \Omega, \nu - \left\lceil\frac{\Omega}{2}\right\rceil\right)G(\mathbf{k}-\mathbf{Q}; \nu-\Omega)\,, \tag{36a}$$

$$\Sigma(\mathbf{k}; \nu) = \sum_{\mathbf{Q},\Omega}\left[w^D(\mathbf{Q};\Omega)\lambda^D\left(\mathbf{Q}, \mathbf{k}-\mathbf{Q}; \Omega, \nu - \left\lceil\frac{\Omega}{2}\right\rceil\right) - 2U^D(\mathbf{Q}, \mathbf{k};\Omega, \nu)\right]G(\mathbf{k}-\mathbf{Q}; \nu-\Omega)\,, \tag{36b}$$

$$\Sigma(\mathbf{k}; \nu) = -\sum_{\mathbf{Q},\Omega} w^{SC}(\mathbf{Q};\Omega)\lambda^{SC}\left(\mathbf{Q}, \mathbf{Q}-\mathbf{k}; \Omega, \left\lceil\frac{\Omega}{2}\right\rceil - \nu\right)G(\mathbf{Q}-\mathbf{k}; \Omega-\nu)\,, \tag{36c}$$

where $U^D(\mathbf{Q}, \mathbf{k};\Omega, \nu) \equiv U^D(k-Q, k|k, k-Q)$. Together with the forms in diagrammatic channels (25), the above equations represent the main result of the present paper.

## 2.5 Expansion in form factors

We now address the possible problems associated with TU solvers that use a truncated form-factor expansion for the fermionic momenta, as the TU fRG [39, 40, 43] and the TU parquet equations [4, 44].

"Truncated unity" refers to the insertion of the unity

$$\mathbb{1} = \int d\mathbf{p}'\delta(\mathbf{p}-\mathbf{p}') = \int d\mathbf{p}'\sum_m f^*_m(\mathbf{p})f_m(\mathbf{p}')\,,$$

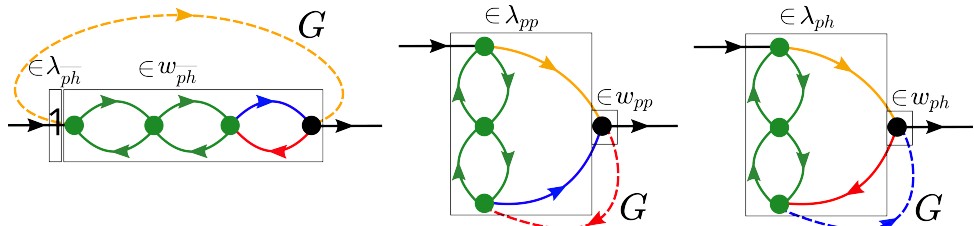

Figure 2: The same self-energy diagram drawn as $\lambda_{\overline{ph}} w_{\overline{ph}} G$ (left), as $\lambda_{pp} w_{pp} G$ (center), and as $\lambda_{ph} w_{ph} G$ (on the right). The dashed line indicates the closing Green's function. Using only an *s*-wave form factor is exact for the computation via $\lambda_{\overline{ph}} w_{\overline{ph}} G$, but not for $\lambda_{pp/ph} w_{pp/ph} G$, due to the information loss induced by the form-factor projections.

and the subsequent truncation to only few form factors in practical applications. For this, we rewrite the above SDE in diagrammatic channels, Eqs. (25), in form-factor notation (analogous arguments hold for the physical channels)

$$\Sigma^{\uparrow}(\mathbf{k}; \nu) = \sum_{\mathbf{Q},\Omega} G^{\downarrow}(\mathbf{k}-\mathbf{Q}; \nu-\Omega) w_{\overline{ph}}^{\widehat{\uparrow\downarrow}}(\mathbf{Q};\Omega) \left[ \sum_m f_m(\mathbf{k}-\mathbf{Q}) \lambda_{\overline{ph};m}^{\widehat{\uparrow\downarrow}} \left( \mathbf{Q};\Omega, \nu - \left\lceil \frac{\Omega}{2} \right\rceil \right) \right], \tag{37a}$$

$$\Sigma^{\uparrow}(\mathbf{k}; \nu) = -\sum_{\mathbf{Q},\Omega} G^{\downarrow}(\mathbf{k}-\mathbf{Q}; \nu-\Omega) w_{ph}^{\uparrow\downarrow}(\mathbf{Q};\Omega) \left[ \sum_m f_m(\mathbf{k}-\mathbf{Q}) \lambda_{ph;m}^{\uparrow\downarrow} \left( \mathbf{Q};\Omega, \nu - \left\lceil \frac{\Omega}{2} \right\rceil \right) \right], \tag{37b}$$

$$\Sigma^{\uparrow}(\mathbf{k}; \nu) = -\sum_{\mathbf{Q},\Omega} G^{\downarrow}(\mathbf{Q}-\mathbf{k}; \Omega-\nu) w_{pp}^{\uparrow\downarrow}(\mathbf{Q};\Omega) \left[ 2\sum_m f_m(\mathbf{Q}-\mathbf{k}) \lambda_{pp;m}^{\uparrow\downarrow} \left( \mathbf{Q};\Omega, \left\lceil \frac{\Omega}{2} \right\rceil - \nu \right) - 1 \right], \tag{37c}$$

where $\{f_m(\mathbf{k})\}_{m=0}^{\infty}$ is a set of form factors defined on the Brillouin zone. The range of their real space representation is determined by the bond length. If the results are converged in the number of form factors, all three single-boson exchange expressions of the SDE (37) yield the same result. This is in general not the case if only a small number is considered. In fact, the restriction to a small number of form factors leads in general to a violation of the crossing symmetry [44]. In particular, a truncation in the form factors fully includes the diagrams reducible in the corresponding channel (any *r*-reducible diagram in the formulation including $\lambda_r$), but only partially those reducible in the other channels. This is exemplified in Fig. 2: using only an *s*-wave form factor; i.e., restricting to $f_0(\mathbf{k}) = 1$, the diagram shown in the figure is computed exactly in the $\overline{ph}$ formulation of the SDE. Indeed, the argument of the bosonic propagator *w*, being entirely bosonic, is not affected by the *s*-wave form-factor truncation. However, the same diagram is not accounted for correctly in its formulation, since $\lambda_{ph/pp}$ depends also on a fermionic argument which is not captured by the constant *s*-wave form factor. We note that all ladder diagrams formulated in the corresponding channel are treated exactly (see left diagram of Fig. 2 as an example), only the corrections from the other channels to these ladders are affected by the truncation in the number of considered form factors.

To summarize, some diagrams are not treated optimally in the single-boson exchange SDE with respect to a truncation in form factors. In this case, the computation of the self-energy generally depends on the choice of the channel, as will be shown in the application to the 2D Hubbard model presented in the next section. Specifically, when the fermionic momentum dependence is expanded in form factors, the crossing symmetry between the particle-hole channels is broken [12, 44].

# 3 Application to the fRG: The pseudogap opening in the 2D Hubbard model

We now apply the SDE in the single-boson exchange representation to the fRG [45, 46]. We focus on the pseudogap opening in the 2D Hubbard model at weak coupling,[2] where forefront algorithmic advancements brought the fRG to a quantitatively reliable level [51, 52]. In particular, the multiloop extension [53, 54] allows one to recover the parquet approximation [31, 55, 56]. In this scheme, the self-energy flow is determined by the derivative of the SDE. In the implementation based on the parquet decomposition, the use of the SDE has been shown to be crucial for detecting the pseudogap opening. [47]. Here, we employ the single-boson exchange formulation of the SDE derived above, extending the single-boson exchange formulation of the fRG [10, 15, 57] to the computation of the self-energy.

For the Hubbard model [58] with nearest-neighbor hopping amplitude $t$, chemical potential $\mu$, and local Coulomb repulsion $U$, the classical action is of the form (1), with

$$U_{\sigma_{1'}\sigma_{2'}|\sigma_1\sigma_2}(k_{1'},k_{2'}|k_1,k_2) = -U\delta(k_{1'}+k_{2'}-k_1-k_2)(\delta_{\sigma_{1'},\sigma_1}\delta_{\sigma_{2'},\sigma_2} - \delta_{\sigma_{1'},\sigma_2}\delta_{\sigma_{2'},\sigma_1})$$
$$\times (1-\delta_{\sigma_1,\sigma_2}),\qquad(38)$$

and the bare propagator given by

$$G^{-1}_{0,\sigma_{1'}|\sigma_1}(k_{1'}|k_1) = (i\nu_1 - \epsilon_{\mathbf{k}_1} + \mu)\delta(k_{1'}-k_1),\qquad(39)$$

where the dispersion relation reads $\epsilon_{\mathbf{k}} = -2t[\cos(k_x) + \cos(k_y)]$. Throughout our analysis, we consider $t \equiv 1$ as energy unit and focus on $|U| = 2$ and half filling with $\langle \hat{n} \rangle = 1$. Using the $T$ flow [59], allows us to track the temperature evolution of the pseudogap opening along the renormalization-group flow. The flow equations for the bosonic propagator, the fermion-boson coupling, and the rest function are reported in Appendix D. The self-energy flow is determined from the derivative of the SDE (31), (32), and (34). For the magnetic channel formulation, Eq. (31) leads to

$$\dot{\Sigma} = \dot{G} \cdot \left(w^{\mathrm{M}}\lambda^{\mathrm{M}}\right) + G \cdot \left(\dot{w}^{\mathrm{M}}\lambda^{\mathrm{M}}\right) + G \cdot \left(w^{\mathrm{M}}\dot{\lambda}^{\mathrm{M}}\right).\qquad(40)$$

The momentum and frequency dependencies are obtained by following the explicit form (37a). The corresponding expressions for the D and SC channels can be derived analogously. We note that the derivative of the self-energy appearing in the Katanin correction for $\dot{G}$ on the right-hand side is replaced by the conventional $1\ell$ flow. In order to account for the full feedback, the equation should be iterated until convergence. Since this results only in quantitative corrections [52], we neglect the iterations here.

We here perform a two-loop ($2\ell$) computation[3] that neglects the flow of the $U$-irreducible rest functions (in the considered parameter regime its effects are very small [15]). Specifically, we use $n = 8$ positive fermionic and $2n$ bosonic frequencies for the parametrization of the fermion-boson coupling and rest function, whereas for the bosonic propagators we use $64n$ positive bosonic frequencies. For the self-energy, we use $10n$ positive fermionic frequencies, and for the bubble integrand we use $64n$ positive bosonic and $64n$ positive fermionic frequencies. The fermionic momentum dependence of the fermion-boson coupling is accounted for by a form-factor expansion, where we consider only the local $s$-wave contribution since at half filling the physics is dominated by antiferromagnetic fluctuations. For the transfer momentum parametrization, in addition to $16 \times 16$ momentum patches distributed on an equally spaced grid in the Brillouin zone, we take into account a finer grid around the antiferromagnetic peak

---

[2]See Ref. [50] for a review.

[3]Differently to the conventional $1\ell$ scheme, the $2\ell$ truncation is exact to third order in $U$ with corrections of $\mathcal{O}(U^4)$. For the details on the implementation in the single-boson exchange formulation, we refer to Ref. [60].

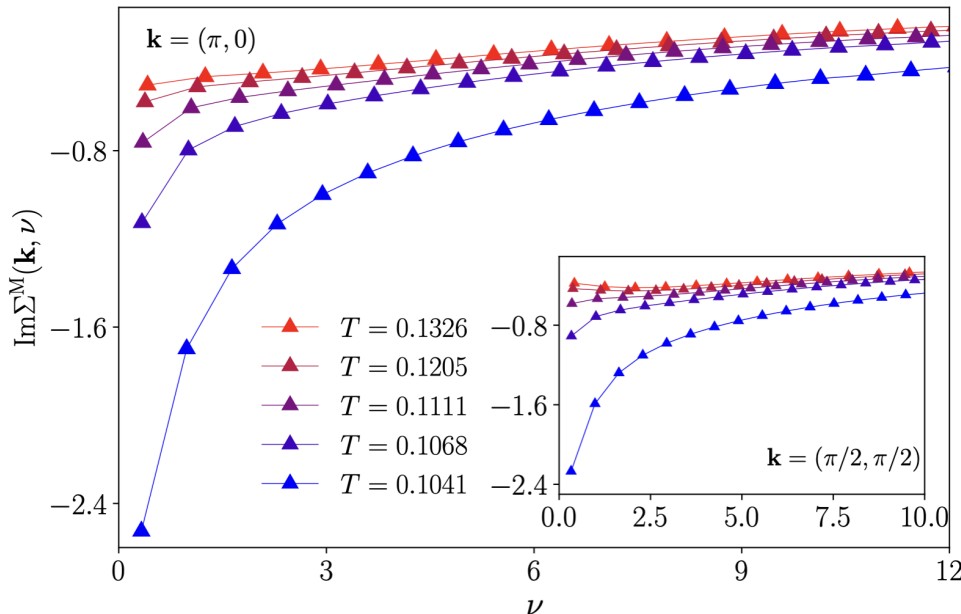

Figure 3: Imaginary part of the self-energy as a function of Matsubara frequencies at half filling ($\mu = 0$), $U = 2$, and various temperatures, as determined by its expression in the magnetic channel (36a). At the antinodal point $\mathbf{k} = (\pi, 0)$ displayed in the main panel, the pseudogap opens at higher temperatures as compared to the nodal point $\mathbf{k} = (\pi/2, \pi/2)$, see inset.

at $\mathbf{k} = (\pi, \pi)$ and the superconducting peak at $\mathbf{k} = (0,0)$. The bubble transfer momentum dependence is calculated on a much denser grid of $80 \times 80$ momentum patches, see Ref. [57] for the details.

We here focus on the analysis in Matsubara frequencies. A non-Fermi-liquid behavior can be signaled by deviations of the quasiparticle weight

$$Z(\mathbf{k}) = \left(1 - \frac{\partial \mathrm{Re}\Sigma(\nu, \mathbf{k})}{\partial \nu}\bigg|_{\nu \to 0}\right)^{-1} < 1, \tag{41}$$

where $\nu$ is a real frequency. In the limit of low temperatures, $\partial_\nu \mathrm{Re}\Sigma(\nu, \mathbf{k})|_{\nu \to 0}$ can be translated to Matsubara frequencies. The gap opening can then be observed directly in the imaginary part of the self-energy bending towards negative large values. In contrast, the Fermi-liquid regime is always characterized by a bending towards small values. In Figs. 3 and 4 we present the fRG results obtained for the different channel representations of the SDE. Due to their equivalence, these are expected to yield the same result. In the TU-fRG, however, the pseudogap opening is only observed in the magnetic channel representation and not in the density or superconducting one. As we will discuss below, this is a consequence of the reduced number of form factors and their convergence, which is different in the three channel representations. We first focus on the magnetic (or $\overline{ph}$) channel data shown in Fig. 3. At low temperatures, we observe an insulating behavior initially at the antinodal point $\mathbf{k} = (0, \pi)$. At the nodal point $\mathbf{k} = (\pi/2, \pi/2)$, the gap opening occurs at lower temperatures. These findings agree with the results obtained with the parquet formulation [47]. The results for the density and superconducting channel representation of the SDE are reported in Fig. 4. We find equal self-energies in the superconducting and density formulations, in agreement to the expectation based on $\mathrm{SU}_P(2)$-symmetry on the square lattice at half-filling. Differently from the magnetic channel, these representations fail to capture the pseudogap opening even at the antinodal point, where it should be more pronounced. Note also the different scales with respect to Fig. 3. This behavior can be

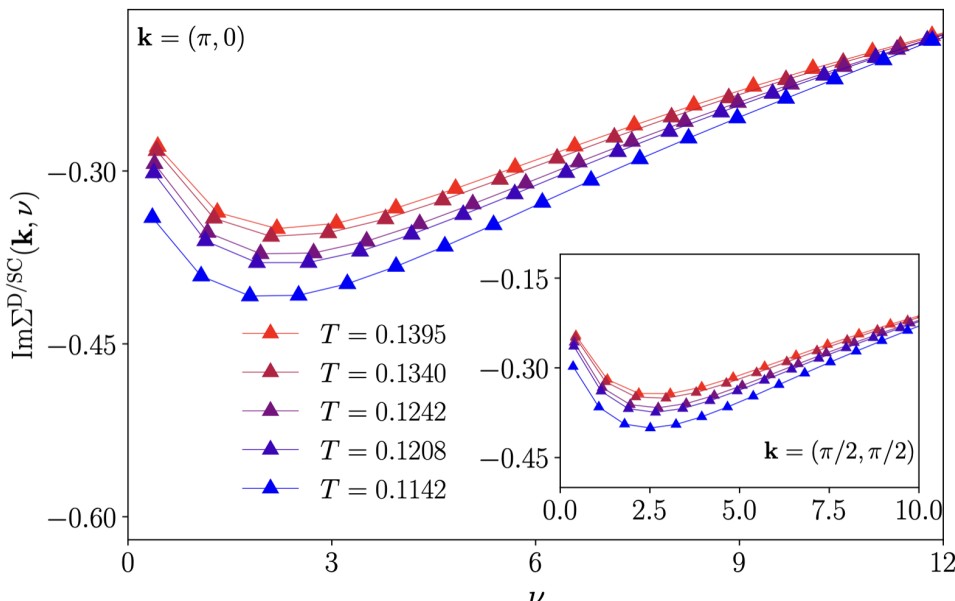

Figure 4: Same as Fig. 3, but determined by the density and superconducting channel using Eqs. (36b) and (36c). It can be clearly seen that these representations fail to capture the pseudogap opening both at the antinodal and the nodal point. Note also the different scales with respect to Fig. 3.

understood in the light of the discussion in Section 2.5: The magnetic fluctuations driving the pseudogap opening are not translated efficiently to the subleading channels in the TU fRG and the flow diverges before the onset of the pseudogap opening develops. As a consequence, the self-energy retains a Fermi-liquid nature for all values of the temperature in our analysis. We note that the pseudo-critical transition temperature in the density and superconducting channel representations appears to be higher than for magnetic one. This is due to the information loss induced by the form-factor projections which reduces the screening of the strong antiferromagnetic fluctuations at half filling. In particular, the two channel representations appear to be affected in the same way, see also Fig. 2. At finite doping, we expect the same qualitative behavior since the pseudogap opening is driven by antiferromagnetic fluctuations also in this case [61, 62].

We finally note that the dependence on the different representations is due to the different convergence in the number of form factors. In the magnetic channel, the pseudogap opening is captured already by the single $s$-wave form factor considered here, while in the density and superconducting ones it is insufficient. This problem can be circumvented by using the parquet-based formulation of the SDE. The latter does not induce a bias between the different physical channels and captures the pseudogap opening within the $s$-wave truncation [47]. In this formulation, replacing the two-particle vertex by its single-boson exchange representation, the SDE includes also multiboson contributions [63].

We note that the self-energy is independent of the sign of $U$ [64]. Moreover, at half filling, the Shiba transformation [65] maps the attractive ($U < 0$) to the repulsive Hubbard model. Specifically, the $s$-wave superconducting fluctuations at $\mathbf{Q} = (0, 0)$ and the density fluctuations at $\mathbf{Q} = (\pi, \pi)$ in the attractive model correspond to the antiferromagnetic spin components in the repulsive model. Consequently, as expected, for the attractive Hubbard, we obtain the same results, but with exchanged channels: the dominant density and superconducting fluctuations drive the pseudogap opening observed in the corresponding channel representations, while no pseudogap opening is detected in the magnetic channel representation. At half filling,

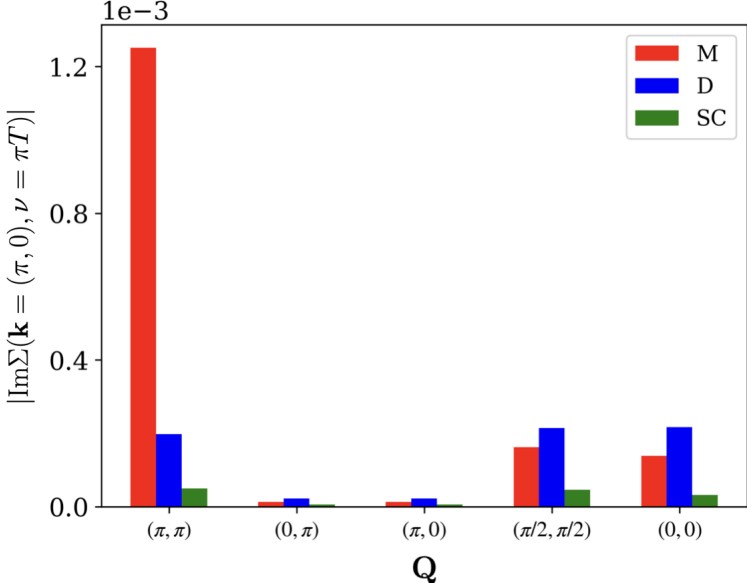

Figure 5: Fluctuation diagnostics of the imaginary part of the self-energy at the antinodal point, for the repulsive Hubbard model at half filling, $U = 2$ and $T = 0.13$. The histogram bars display the contributions of the different bosonic momenta $Q = (\mathbf{Q}, \Omega = 0)$ in the magnetic (red), density (blue) and superconducting (green) representations. The pronounced red bar at $\mathbf{Q} = (\pi, \pi)$ clearly shows the dominant contributions of the antiferromagnetic spin fluctuations.

the results obtained from the density and superconducting channel representations for the attractive model coincide with the ones determined by the magnetic channel in the repulsive model and the magnetic channel representation results for the attractive model with the ones determined by the density and superconducting channels in the repulsive model. Also in this case, the channels controlling the physical behavior yield the correct description.

A more detailed understanding can be obtained by applying the fluctuation diagnostics approach [34–38] to analyze the main collective mode contributions to the self-energy in both the repulsive and attractive cases. We recall that the single-boson exchange SDE for the self-energy in the different channels, Eqs. (36), includes – by construction – an integral over processes in which the Green's function is renormalized by a momentum and frequency dependent boson as well as by a fermion-boson coupling. Although, in general, all momenta and frequencies will contribute, in the representation reflecting the physically relevant fluctuations, specific momenta and frequencies will dominate the contributions to the integral. In the framework of the fluctuation diagnostics, this indicates that a boson of the corresponding channel can be deemed primarily responsible for the self-energy/spectral feature under investigation. For our analysis of the pseudogap opening, following Refs. [34, 38] we focus on the first Matsubara frequency at the antinodal point $\mathbf{k} = (0, \pi)$. The corresponding fluctuation diagnostics results for the formulation of the self-energy in the magnetic, density, and superconducting channel are reported in Fig. 5 for the repulsive Hubbard model. In particular, we visualize the integrands of Eqs. (36) as a function of the bosonic transfer momentum $\mathbf{Q}$ (and $\Omega = 0$), since this vector defines the transfer momentum of the corresponding collective modes. Then, a dominant contribution appearing as a peak in the integrand of the magnetic or charge representation of the self-energy at $\mathbf{Q} = (\pi, \pi)$ can be attributed to antiferromagnetic or charge density wave fluctuations respectively, while a peak in the superconducting representation of the SDE at $\mathbf{Q} = (0, 0)$ hints at strong pairing fluctuations. The data in Fig. 5 shed light on the underlying physics of the pseudogap observed in the fRG data: In the magnetic representa-

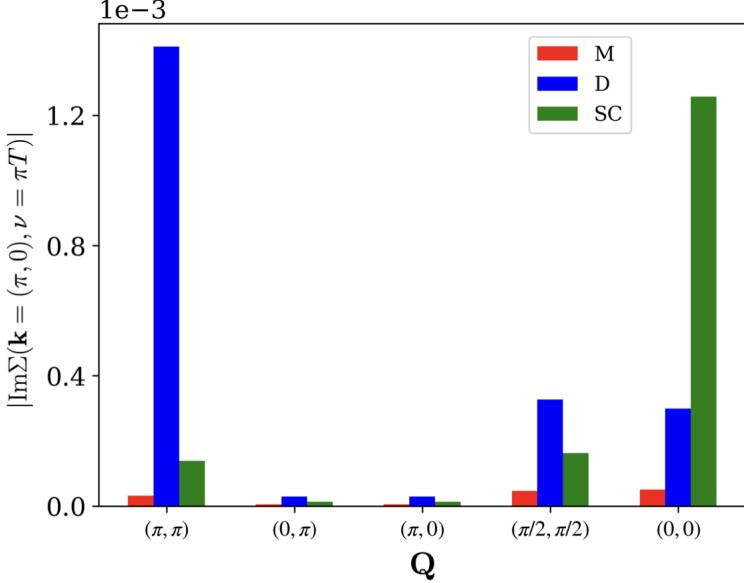

Figure 6: Same as Fig. 5, but for the attractive Hubbard model ($U = -2$). In this case, the density and superconducting fluctuations at $\mathbf{Q} = (\pi, \pi)$ and respectively $\mathbf{Q} = (0, 0)$ dominate.

tion, the dominant contribution at $\mathbf{Q} = (\pi, \pi)$ reflects the strong influence of antiferromagnetic fluctuations, while the density and superconducting representations yield an essentially featureless momentum distribution, not presenting significant contributions to the self-energy for any specific momentum vector.

Reversing the sign of the interaction $U$ in our model, we carry out an analogous analysis to characterize the physics underlying the pseudogap opening in the attractive Hubbard model. Here, the fluctuation diagnostics identifies charge density wave and $s$-wave pairing fluctuations as key players, see Fig. 6. The results show significant contributions at $\mathbf{Q} = (\pi, \pi)$ in the density representation and at $\mathbf{Q} = (0, 0)$ in the superconducting one. At the same time, now the magnetic representation does not display any pronounced momentum-selective behavior. We note that the displayed results include only the $\Omega = 0$ evaluation of the integrand, from which the degeneracy of the density and superconducting contributions can not be directly inferred (the same applies to Fig. 5).

## 4  Conclusions and outlook

We derived the expression for the Schwinger–Dyson equation (SDE) for the self-energy in the single-boson exchange formulation. The employed formalism makes use of matrices to encode the spin structure and allows for a compact representation of the SDE. The resulting equation exhibits a simple form involving only the bosonic propagator and the fermion-boson vertex (and not the rest function). Moreover, the single-boson exchange SDE is a one-loop equation, in contrast to the two-loop nature of its conventional expression. As a result of the symmetry of the systems (e.g. SU(2), U(1), ...) our SDE expression can be recast in several, formally equivalent representations, which essentially corresponds to the physical scattering channels of the system.

However, such a formal equivalence is generally broken if truncated-unity (TU) approximations are included in the algorithm used for the calculations (e.g., for parquet and fRG implementations using a restricted number of form factors). In particular, the information loss introduced by the projection of the momentum dependence directly affects specific channel representations of the SDE and may be reflected in an unphysical dependence on the chosen SDE form. In the specific case of the fRG presented in this work, we analyzed the pseudogap opening in the 2D Hubbard model at half filling. We found that the self-energy flow yields the expected behavior already by the *s*-wave form factor in the magnetic channel representation. In contrast, the convergence in the number of form factors in the density and superconducting channel representation is slower. For these, an *s*-wave computation does not provide an accurate description of the antiferromagnetic fluctuations dominating the physics.

As an outlook, the extension to non-local interactions, only briefly alluded to here, represents an important step / plays an important role also in the generalization of the fluctuations diagnostics [34, 37, 66] as a versatile post-processing tool to quantify the contributions of the different scattering processes. Concerning the fRG implementation, further developments include the extension to the strong coupling regime by the combination with the dynamical mean-field theory (DMFT) [67, 68] in the so-called DMF$^2$RG [10, 14, 69, 70].

## Acknowledgments

The authors thank L. Del Re, D. Kiese, M. Krämer, J. von Delft for valuable discussions and insights and F. Krien for his comments on the manuscript.

**Funding information**   We acknowledge financial support from the Deutsche Forschungsgemeinschaft (DFG) within the research unit FOR5413 (Grant No. 465199066). This research was also funded in part by the Austrian Science Fund (FWF) 10.55776/I6946 and 10.55776/I5868 (as a part of the research unit "QUAST" FOR 5249 of the DFG). The calculations have been performed on the Vienna Scientific Cluster (VSC). M. G. acknowledges funding from the International Max Planck Research School for Quantum Science and Technology (IMPRS-QST) and P. B. support by the German National Academy of Sciences Leopoldina through Grant No. LPDS 2023-06 and funding from U.S. National Science Foundation grant No. DMR2245246.

**Author contributions**   S. A., D. V., and A. T. proposed and coordinated the project, supervising the theoretical work and the analysis of the numerical results. M. P. derived the analytical results with input from M. G., K. F., and S. H.. M. P. and K. F. carried out the fRG calculations with the codes developed by P. B. and D. V. and by S. H., A. A.-E., and K. F.. M. P. analyzed the numerical results together with K. F. and A. A.-E.. M. P. prepared the figures, and M. P, M. G, K. F., and S. A. wrote the paper with input from all authors.

## A   Details on the formalism

In this appendix, we present the notation to handle the spin and momentum/frequency structure of the single-boson exchange vertices introduced in Ref. [9] and discussed in more detail in Ref. [71].

## A.1 Matrix representation of the spin structure

The summation over spin indices for products of four-point objects such as $\Pi$ and $V$ or any other object with the same index structure can be carried out efficiently by storing their spin components in $4 \times 4$ matrices. The summation over spin indices is then carried out by performing standard matrix products. Assuming that $A = \Pi, V$, etc., the matrices in the different diagrammatic channels read

$$
A_{ph} = \begin{bmatrix} A_{ph}^{\widehat{\uparrow\downarrow}} & 0 & 0 & 0 \\ 0 & A_{ph}^{\widehat{\downarrow\uparrow}} & 0 & 0 \\ 0 & 0 & A_{ph}^{\uparrow\uparrow} & A_{ph}^{\downarrow\uparrow} \\ 0 & 0 & A_{ph}^{\uparrow\downarrow} & A_{ph}^{\downarrow\downarrow} \end{bmatrix}, \quad A_{\overline{ph}} = \begin{bmatrix} A_{\overline{ph}}^{\uparrow\downarrow} & 0 & 0 & 0 \\ 0 & A_{\overline{ph}}^{\downarrow\uparrow} & 0 & 0 \\ 0 & 0 & A_{\overline{ph}}^{\uparrow\uparrow} & A_{\overline{ph}}^{\widehat{\uparrow\downarrow}} \\ 0 & 0 & A_{\overline{ph}}^{\widehat{\downarrow\uparrow}} & A_{\overline{ph}}^{\downarrow\downarrow} \end{bmatrix}, \quad A_{pp} = \begin{bmatrix} A_{pp}^{\uparrow\uparrow} & 0 & 0 & 0 \\ 0 & A_{pp}^{\downarrow\downarrow} & 0 & 0 \\ 0 & 0 & A_{pp}^{\uparrow\downarrow} & A_{pp}^{\widehat{\uparrow\downarrow}} \\ 0 & 0 & A_{pp}^{\downarrow\uparrow} & A_{pp}^{\widehat{\downarrow\uparrow}} \end{bmatrix}.
$$
(A.1a)

Following the definition of the bubble products in Eqs. (6), the products involving these objects are obtained through usual matrix multiplications. There is always a "natural" spin component where the multiplication has a diagonal structure, i.e., $A_{ph}^{\widehat{\uparrow\downarrow}}$, $A_{\overline{ph}}^{\uparrow\downarrow}$ and $A_{pp}^{\uparrow\uparrow}$ (and $A_{ph}^{\widehat{\downarrow\uparrow}}$, $A_{\overline{ph}}^{\downarrow\uparrow}$, $A_{pp}^{\downarrow\downarrow}$). For the other spin components, the multiplication is non-diagonal. Explicitly:

$$
[A_{ph} \circ B_{ph}]^{\widehat{\uparrow\downarrow}} = A_{ph}^{\widehat{\uparrow\downarrow}} \circ B_{ph}^{\widehat{\uparrow\downarrow}}, \qquad [A_{ph} \circ B_{ph}]^{\widehat{\downarrow\uparrow}} = A_{ph}^{\widehat{\downarrow\uparrow}} \circ B_{ph}^{\widehat{\downarrow\uparrow}},
$$

$$
\begin{bmatrix} [A_{ph} \circ B_{ph}]^{\uparrow\uparrow} & [A_{ph} \circ B_{ph}]^{\downarrow\uparrow} \\ [A_{ph} \circ B_{ph}]^{\uparrow\downarrow} & [A_{ph} \circ B_{ph}]^{\downarrow\downarrow} \end{bmatrix} = \begin{bmatrix} A_{ph}^{\uparrow\uparrow} & A_{ph}^{\downarrow\uparrow} \\ A_{ph}^{\uparrow\downarrow} & A_{ph}^{\downarrow\downarrow} \end{bmatrix} \circ \begin{bmatrix} B_{ph}^{\uparrow\uparrow} & B_{ph}^{\downarrow\uparrow} \\ B_{ph}^{\uparrow\downarrow} & B_{ph}^{\downarrow\downarrow} \end{bmatrix},
$$
(A.2a)

$$
[A_{\overline{ph}} \circ B_{\overline{ph}}]^{\uparrow\downarrow} = A_{\overline{ph}}^{\uparrow\downarrow} \circ B_{\overline{ph}}^{\uparrow\downarrow}, \qquad [A_{\overline{ph}} \circ B_{\overline{ph}}]^{\downarrow\uparrow} = A_{\overline{ph}}^{\downarrow\uparrow} \circ B_{\overline{ph}}^{\downarrow\uparrow},
$$

$$
\begin{bmatrix} [A_{\overline{ph}} \circ B_{\overline{ph}}]^{\uparrow\uparrow} & [A_{\overline{ph}} \circ B_{\overline{ph}}]^{\widehat{\uparrow\downarrow}} \\ [A_{\overline{ph}} \circ B_{\overline{ph}}]^{\widehat{\downarrow\uparrow}} & [A_{\overline{ph}} \circ B_{\overline{ph}}]^{\downarrow\downarrow} \end{bmatrix} = \begin{bmatrix} A_{\overline{ph}}^{\uparrow\uparrow} & A_{\overline{ph}}^{\widehat{\uparrow\downarrow}} \\ A_{\overline{ph}}^{\widehat{\downarrow\uparrow}} & A_{\overline{ph}}^{\downarrow\downarrow} \end{bmatrix} \circ \begin{bmatrix} B_{\overline{ph}}^{\uparrow\uparrow} & B_{\overline{ph}}^{\widehat{\uparrow\downarrow}} \\ B_{\overline{ph}}^{\widehat{\downarrow\uparrow}} & B_{\overline{ph}}^{\downarrow\downarrow} \end{bmatrix},
$$
(A.2b)

$$
[A_{pp} \circ B_{pp}]^{\uparrow\uparrow} = A_{ph}^{\uparrow\uparrow} \circ B_{ph}^{\uparrow\uparrow}, \qquad [A_{pp} \circ B_{pp}]^{\downarrow\downarrow} = A_{pp}^{\downarrow\downarrow} \circ B_{pp}^{\downarrow\downarrow},
$$

$$
\begin{bmatrix} [A_{pp} \circ B_{pp}]^{\uparrow\downarrow} & [A_{pp} \circ B_{pp}]^{\widehat{\uparrow\downarrow}} \\ [A_{pp} \circ B_{pp}]^{\widehat{\downarrow\uparrow}} & [A_{pp} \circ B_{pp}]^{\downarrow\uparrow} \end{bmatrix} = \begin{bmatrix} A_{pp}^{\uparrow\downarrow} & A_{pp}^{\widehat{\uparrow\downarrow}} \\ A_{pp}^{\downarrow\uparrow} & A_{pp}^{\widehat{\downarrow\uparrow}} \end{bmatrix} \circ \begin{bmatrix} B_{pp}^{\uparrow\downarrow} & B_{pp}^{\widehat{\uparrow\downarrow}} \\ B_{pp}^{\downarrow\uparrow} & B_{pp}^{\widehat{\downarrow\uparrow}} \end{bmatrix}.
$$
(A.2c)

Note that the products of $U$-reducible vertices $\bar{\lambda}_r \bullet w_r$ and $w_r \bullet \lambda_r$ exactly follow that structure. Also the spin structure of the triple products $V \circ \Pi_r \circ U$ and $U \circ \Pi_r \circ V$ are obtained by applying the matrix products twice. We also stress that, as in the main text, the involved summations over frequencies and momenta are not accounted for and still have to be considered.

Making use of channel-dependent momentum/frequency parametrization (cf. Fig. 7) and of the crossing symmetries

$$
V_{12|34} = -V_{21|34} = -V_{12|43} = V_{21|43},
$$
(A.3)

one can deduce the following relations for the vertex:

$$
\begin{aligned}
V_{ph}^{\uparrow\downarrow}(Q_{ph}, k_{ph}, k'_{ph}) &= -V_{\overline{ph}}^{\widehat{\uparrow\downarrow}}(-\mathbf{Q}_{ph}, \mathbf{Q}_{ph} + \mathbf{k}'_{ph}, \mathbf{Q}_{ph} + \mathbf{k}_{ph}, -\Omega_{ph}, \nu'_{ph}, \nu_{ph}) \\
&= -V_{\overline{ph}}^{\widehat{\downarrow\uparrow}}(\mathbf{Q}_{ph}, \mathbf{k}_{ph}, \mathbf{k}'_{ph}, \Omega_{ph}, \nu_{ph}, \nu'_{ph}) \\
&= V_{ph}^{\downarrow\uparrow}(-\mathbf{Q}_{ph}, \mathbf{Q}_{ph} + \mathbf{k}'_{ph}, \mathbf{Q}_{ph} + \mathbf{k}_{ph}, -\Omega_{ph}, \nu'_{ph}, \nu_{ph}),
\end{aligned}
$$
(A.4a)

$$V_{\overline{ph}}^{\uparrow\downarrow}(Q_{\overline{ph}}, k_{\overline{ph}}, k'_{\overline{ph}}) = -V_{\overline{ph}}^{\widehat{\uparrow\downarrow}}(-\mathbf{Q}_{\overline{ph}}, \mathbf{Q}_{\overline{ph}} + \mathbf{k}'_{\overline{ph}}, \mathbf{Q}_{\overline{ph}} + \mathbf{k}_{\overline{ph}}, -\Omega_{\overline{ph}}, \nu'_{\overline{ph}}, \nu_{\overline{ph}})$$

$$= -V_{\overline{ph}}^{\widehat{\downarrow\uparrow}}(\mathbf{Q}_{\overline{ph}}, \mathbf{k}_{\overline{ph}}, \mathbf{k}'_{\overline{ph}}, \Omega_{\overline{ph}}, \nu_{\overline{ph}}, \nu'_{\overline{ph}})$$

$$= V_{\overline{ph}}^{\downarrow\uparrow}(-\mathbf{Q}_{\overline{ph}}, \mathbf{Q}_{\overline{ph}} + \mathbf{k}'_{\overline{ph}}, \mathbf{Q}_{\overline{ph}} + \mathbf{k}_{\overline{ph}}, -\Omega_{\overline{ph}}, \nu'_{\overline{ph}}, \nu_{\overline{ph}}), \qquad \text{(A.4b)}$$

$$V_{pp}^{\uparrow\downarrow}(Q_{pp}, k_{pp}, k'_{pp}) = -V_{pp}^{\widehat{\uparrow\downarrow}}(\mathbf{Q}_{pp}, \mathbf{Q}_{pp} - \mathbf{k}_{pp}, \mathbf{k}'_{pp}, \Omega_{pp}, -\nu_{pp} + \delta\Omega_{pp}, \nu'_{pp})$$

$$= -V_{pp}^{\widehat{\downarrow\uparrow}}(\mathbf{Q}_{pp}, \mathbf{k}_{pp}, \mathbf{Q}_{pp} - \mathbf{k}'_{pp}, \Omega_{pp}, \nu_{pp}, -\nu'_{pp} + \delta\Omega_{pp})$$

$$= V_{pp}^{\downarrow\uparrow}(\mathbf{Q}_{pp}, \mathbf{Q}_{pp} - \mathbf{k}_{pp}, \mathbf{Q}_{pp} - \mathbf{k}'_{pp}, \Omega_{pp}, -\nu_{pp} + \delta\Omega_{pp}, -\nu'_{pp} + \delta\Omega_{pp}), \quad \text{(A.4c)}$$

where $Q_r = (\mathbf{Q}_r, \Omega_r)$, $k_r = (\mathbf{k}_r, \nu_r)$ and $k'_r = (\mathbf{k}'_r, \nu'_r)$ are the bosonic and fermionic quadrivectors. For convenience, we also defined $\delta\Omega_r = \lceil \frac{\Omega_r}{2} \rceil - \lfloor \frac{\Omega_r}{2} \rfloor$. Note that, since we use symmetrized frequencies, the aforementioned objects depend on $\Omega_r$ through the terms $\lceil \frac{\Omega_r}{2} \rceil$ and $\lfloor \frac{\Omega_r}{2} \rfloor$, as illustrated in Fig. 7. Therefore, when the frequency changes sign ($\Omega_r \rightarrow -\Omega_r$), the following identities are used:

$$\left\lceil -\frac{\Omega_r}{2} \right\rceil = -\left\lfloor \frac{\Omega_r}{2} \right\rfloor, \qquad \left\lfloor -\frac{\Omega_r}{2} \right\rfloor = -\left\lceil \frac{\Omega_r}{2} \right\rceil. \qquad \text{(A.5)}$$

The crossing symmetries for the bubble operators are deduced in a similar manner:

$$\Pi_{ph}^{\widehat{\uparrow\downarrow}}(\mathbf{Q}_{ph}, \mathbf{k}_{ph}, \Omega_{ph}, \nu_{ph}) = \Pi_{ph}^{\widehat{\downarrow\uparrow}}(-\mathbf{Q}_{ph}, \mathbf{Q}_{ph} + \mathbf{k}_{ph}, -\Omega_{ph}, \nu_{ph}), \qquad \text{(A.6a)}$$

$$\Pi_{\overline{ph}}^{\uparrow\downarrow}(\mathbf{Q}_{\overline{ph}}, \mathbf{k}_{\overline{ph}}, \Omega_{\overline{ph}}, \nu_{\overline{ph}}) = \Pi_{\overline{ph}}^{\downarrow\uparrow}(-\mathbf{Q}_{\overline{ph}}, \mathbf{Q}_{\overline{ph}} + \mathbf{k}_{\overline{ph}}, -\Omega_{\overline{ph}}, \nu_{\overline{ph}}), \qquad \text{(A.6b)}$$

$$\Pi_{pp}^{\uparrow\downarrow}(\mathbf{Q}_{pp}, \mathbf{k}_{pp}, \Omega_{pp}, \nu_{pp}) = \Pi_{pp}^{\downarrow\uparrow}(\mathbf{Q}_{pp}, \mathbf{Q}_{pp} - \mathbf{k}_{pp}, \Omega_{pp}, \delta\Omega_{pp} - \nu_{pp}). \qquad \text{(A.6c)}$$

In the matrix space for spin indices, the bubble operators are all diagonal. In particular, this means that the following components vanish:

$$\Pi_{ph}^{\uparrow\downarrow} = \Pi_{ph}^{\downarrow\uparrow} = 0, \qquad \Pi_{\overline{ph}}^{\widehat{\uparrow\downarrow}} = \Pi_{\overline{ph}}^{\widehat{\downarrow\uparrow}} = 0, \qquad \Pi_{pp}^{\widehat{\uparrow\downarrow}} = \Pi_{pp}^{\widehat{\downarrow\uparrow}} = 0. \qquad \text{(A.7)}$$

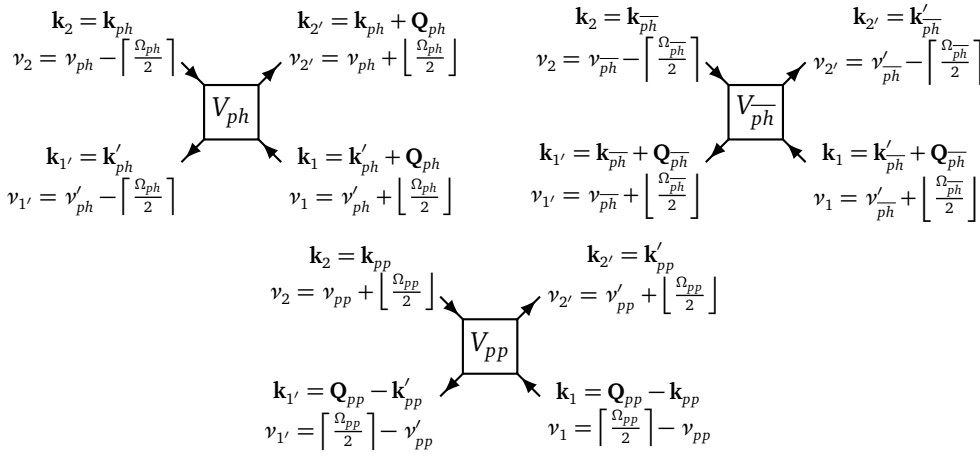

Figure 7: Momentum and frequency conventions for the two-particle vertex in the different channel notations, where $\lceil \cdots \rceil$ ($\lfloor \cdots \rfloor$) rounds the argument up (down) to the nearest bosonic Matsubara frequency. We use a symmetrized notation for the frequencies, which is more convenient for the numerical implementation.

Note that for an SU(2)-symmetric system, the non-vanishing components of the matrices $\mathbf{\Pi}_r$ are all equivalent, since $G^\uparrow = G^\downarrow$. Thus, to define the bubbles in physical channels, it is sufficient to consider the matrix elements of $\Pi_{\overline{ph}}$ for defining $\Pi^M$, the matrix elements of $\Pi_{ph}$ for $\Pi^D$ and the elements of $\Pi_{pp}$ for $\Pi^{SC}$.

We here provide the explicit form for the objects used in the main part of the paper. Recalling the definition of the fermion-boson couplings $\lambda_r = 1_r + 1_r \circ \Pi_r \circ \mathcal{I}_r$ [9], where $\mathcal{I}_r$ is the $U$-irreducible vertex in channel $r$, it is possible to explicitly derive their matrix structure. To simplify the exposition, we will provide the matrix form of the objects $\widetilde{\lambda}_r = \lambda_r - 1_r$, as the corresponding $\lambda_r$ can be easily determined from these. In particular, they read

$$\widetilde{\lambda}_{ph}^{\widehat{\uparrow\downarrow}} = \Pi_{ph}^{\widehat{\uparrow\downarrow}} \circ \mathcal{I}_{ph}^{\widehat{\uparrow\downarrow}},$$

$$\begin{bmatrix} \widetilde{\lambda}_{ph}^{\uparrow\uparrow} & \widetilde{\lambda}_{ph}^{\downarrow\uparrow} \\ \widetilde{\lambda}_{ph}^{\uparrow\downarrow} & \widetilde{\lambda}_{ph}^{\downarrow\downarrow} \end{bmatrix} = \begin{bmatrix} \Pi_{ph}^{\uparrow\uparrow} & 0 \\ 0 & \Pi_{ph}^{\downarrow\downarrow} \end{bmatrix} \circ \begin{bmatrix} \mathcal{I}_{ph}^{\uparrow\uparrow} & \mathcal{I}_{ph}^{\downarrow\uparrow} \\ \mathcal{I}_{ph}^{\uparrow\downarrow} & \mathcal{I}_{ph}^{\downarrow\downarrow} \end{bmatrix} = \begin{bmatrix} \Pi_{ph}^{\uparrow\uparrow} \circ \mathcal{I}_{ph}^{\uparrow\uparrow} & \Pi_{ph}^{\uparrow\uparrow} \circ \mathcal{I}_{ph}^{\downarrow\uparrow} \\ \Pi_{ph}^{\downarrow\downarrow} \circ \mathcal{I}_{ph}^{\uparrow\downarrow} & \Pi_{ph}^{\downarrow\downarrow} \circ \mathcal{I}_{ph}^{\downarrow\downarrow} \end{bmatrix}, \tag{A.8a}$$

$$\widetilde{\lambda}_{\overline{ph}}^{\uparrow\downarrow} = \Pi_{\overline{ph}}^{\uparrow\downarrow} \circ \mathcal{I}_{\overline{ph}}^{\uparrow\downarrow},$$

$$\begin{bmatrix} \widetilde{\lambda}_{\overline{ph}}^{\uparrow\uparrow} & \widetilde{\lambda}_{\overline{ph}}^{\widehat{\uparrow\downarrow}} \\ \widetilde{\lambda}_{\overline{ph}}^{\widehat{\downarrow\uparrow}} & \widetilde{\lambda}_{\overline{ph}}^{\downarrow\downarrow} \end{bmatrix} = \begin{bmatrix} \Pi_{\overline{ph}}^{\uparrow\uparrow} & 0 \\ 0 & \Pi_{\overline{ph}}^{\downarrow\downarrow} \end{bmatrix} \circ \begin{bmatrix} \mathcal{I}_{\overline{ph}}^{\uparrow\uparrow} & \mathcal{I}_{\overline{ph}}^{\widehat{\uparrow\downarrow}} \\ \mathcal{I}_{\overline{ph}}^{\widehat{\downarrow\uparrow}} & \mathcal{I}_{\overline{ph}}^{\downarrow\downarrow} \end{bmatrix} = \begin{bmatrix} \Pi_{\overline{ph}}^{\uparrow\uparrow} \circ \mathcal{I}_{\overline{ph}}^{\uparrow\uparrow} & \Pi_{\overline{ph}}^{\uparrow\uparrow} \circ \mathcal{I}_{\overline{ph}}^{\widehat{\uparrow\downarrow}} \\ \Pi_{\overline{ph}}^{\downarrow\downarrow} \circ \mathcal{I}_{\overline{ph}}^{\widehat{\downarrow\uparrow}} & \Pi_{\overline{ph}}^{\downarrow\downarrow} \circ \mathcal{I}_{\overline{ph}}^{\downarrow\downarrow} \end{bmatrix}, \tag{A.8b}$$

$$\widetilde{\lambda}_{pp}^{\uparrow\uparrow} = 0,$$

$$\begin{bmatrix} \widetilde{\lambda}_{pp}^{\uparrow\downarrow} & \widetilde{\lambda}_{pp}^{\widehat{\uparrow\downarrow}} \\ \widetilde{\lambda}_{pp}^{\widehat{\downarrow\uparrow}} & \widetilde{\lambda}_{pp}^{\downarrow\uparrow} \end{bmatrix} = \begin{bmatrix} \Pi_{pp}^{\uparrow\downarrow} & 0 \\ 0 & \Pi_{pp}^{\downarrow\uparrow} \end{bmatrix} \circ \begin{bmatrix} \mathcal{I}_{pp}^{\uparrow\downarrow} & \mathcal{I}_{pp}^{\widehat{\uparrow\downarrow}} \\ \mathcal{I}_{pp}^{\widehat{\downarrow\uparrow}} & \mathcal{I}_{pp}^{\downarrow\uparrow} \end{bmatrix} = \begin{bmatrix} \Pi_{pp}^{\uparrow\downarrow} \circ \mathcal{I}_{pp}^{\uparrow\downarrow} & \Pi_{pp}^{\uparrow\downarrow} \circ \mathcal{I}^{\widehat{\uparrow\downarrow}} \\ \Pi_{pp}^{\downarrow\uparrow} \circ \mathcal{I}_{pp}^{\widehat{\downarrow\uparrow}} & \Pi_{pp}^{\downarrow\uparrow} \circ \mathcal{I}_{pp}^{\downarrow\uparrow} \end{bmatrix}. \tag{A.8c}$$

The expressions for the other fermion-boson vertex $\bar{\lambda}_r$ are obtained by inverting the order in the multiplication. For the bosonic propagators, only the $pp$ channel presents a different form:

$$w_{pp} = w_{pp}^{\uparrow\downarrow} \begin{bmatrix} 0 & 0 & 0 & 0 \\ 0 & 0 & 0 & 0 \\ 0 & 0 & 1 & -1 \\ 0 & 0 & -1 & 1 \end{bmatrix}. \tag{A.9}$$

As before, this can be derived by exploiting the matrix multiplications involved, recalling the definitions $w_r = U + w_r \bullet P_r \bullet U$, where $P_r = \lambda_r \circ \Pi_r \circ 1_r$ [9]. As the bosonic propagator can be represented as $w_r(Q_r) = U + \lim_{|k_r|,|k_r'| \to \infty} V(Q_r, k_r, k_r')$ [9], the bosonic propagator satisfies the following crossing symmetry based relations: for the $pp$ channel

$$w_{pp}^{\uparrow\downarrow}(Q_{pp}) = -w_{pp}^{\widehat{\uparrow\downarrow}}(Q_{pp}) = -w_{pp}^{\widehat{\downarrow\uparrow}}(Q_{pp}) = w_{pp}^{\downarrow\uparrow}(Q_{pp}), \qquad w_{pp}^{\uparrow\uparrow}(Q_{pp}) = w_{pp}^{\downarrow\downarrow}(Q_{pp}) = 0, \tag{A.10}$$

an similarly in the $ph$ and in the $\overline{ph}$ channels

$$w_{ph}^{\uparrow\downarrow}(Q_{ph}) = -w_{ph}^{\widehat{\downarrow\uparrow}}(Q_{ph}) = w_{ph}^{\downarrow\uparrow}(-Q_{ph}) = -w_{ph}^{\widehat{\uparrow\downarrow}}(-Q_{ph}), \tag{A.11a}$$

$$w_{\overline{ph}}^{\uparrow\downarrow}(Q_{\overline{ph}}) = -w_{\overline{ph}}^{\widehat{\downarrow\uparrow}}(Q_{\overline{ph}}) = w_{\overline{ph}}^{\downarrow\uparrow}(-Q_{\overline{ph}}) = -w_{\overline{ph}}^{\widehat{\uparrow\downarrow}}(-Q_{\overline{ph}}), \tag{A.11b}$$

$$w_{\overline{ph}}^{\uparrow\uparrow/\downarrow\downarrow}(Q_{\overline{ph}}) = -w_{ph}^{\uparrow\uparrow/\downarrow\downarrow}(Q_{\overline{ph}}) = w_{\overline{ph}}^{\uparrow\uparrow/\downarrow\downarrow}(-Q_{\overline{ph}}) = -w_{ph}^{\uparrow\uparrow/\downarrow\downarrow}(-Q_{\overline{ph}}). \tag{A.11c}$$

Moreover, using the definition of the fermion-boson vertex $\bar{\lambda}_r = 1_r + \mathcal{I}_r \circ \Pi_r \circ 1_r$, we find the crossing symmetry for the $pp$ channel

$$\lambda_{pp}^{\widehat{\downarrow\uparrow}}(Q_{pp}, k_{pp}) = \sum_{k''} \Pi_{pp}^{\downarrow\uparrow}(Q_{pp}, k'') \mathcal{I}_{pp}^{\widehat{\downarrow\uparrow}}(Q_{pp}, k_{pp}, k'')$$

$$= -\sum_{k''} \Pi_{pp}^{\uparrow\downarrow}(Q_{pp}, k'') \mathcal{I}_{pp}^{\uparrow\downarrow}(Q_{pp}, k_{pp}, k'') = 1 - \lambda_{pp}^{\uparrow\downarrow}(Q_{pp}, k_{pp}). \tag{A.12}$$

Analogously, for the $ph$ and the $\overline{ph}$ channels, we obtain

$$\lambda_{ph}^{\uparrow\downarrow}(Q_{ph}, k_{ph}) = \lambda_{\overline{ph}}^{\widehat{\downarrow\uparrow}}(Q_{ph}, k_{ph}), \tag{A.13a}$$

$$\lambda_{\overline{ph}}^{\uparrow\downarrow}(Q_{ph}, k_{ph}) = \lambda_{ph}^{\widehat{\downarrow\uparrow}}(Q_{\overline{ph}}, k_{\overline{ph}}), \tag{A.13b}$$

$$\lambda_{\overline{ph}}^{\uparrow\uparrow/\downarrow\downarrow}(Q_{\overline{ph}}, k_{\overline{ph}}) = \lambda_{ph}^{\uparrow\uparrow/\downarrow\downarrow}(Q_{\overline{ph}}, k_{\overline{ph}}). \tag{A.13c}$$

Note that following the crossing-symmetry related Eqs. (A.10) and (A.12) in the $pp$ channel, the matrix representations of $w_{pp}$, $\bar{\lambda}_{pp}$ and $\lambda_{pp}$ are well defined even though some $4 \times 4$ matrices are not invertible in the space of all spin components [71].

## A.2  Momentum and frequency conventions

This section aims to report important relations to extract the momentum and frequency dependence (according to our conventions defined by Fig. 7) for the results obtained with the formalism outlined in the main text.

To begin with, we focus on the first version of the loop product, Eq. (7), involved in various forms of the SDE encountered in this paper

$$[A \cdot G]_{1'|1} = A_{1'2'|12} G_{2|2'} = \sum_{k_2, k_{2'}} A_{\sigma_{1'}\sigma_{2'}|\sigma_1\sigma_2}(k_{1'}, k_{2'}|k_1, k_2) G_{\sigma_2|\sigma_{2'}}(k_2|k_{2'}). \tag{A.14}$$

Assuming translational invariance and energy conservation, we have

$$A_{\sigma_{1'}\sigma_{2'}|\sigma_1\sigma_2}(k_{1'}, k_{2'}|k_1, k_2) = \delta_{k_{1'}+k_{2'}, k_1+k_2} A_{\sigma_{1'}\sigma_{2'}|\sigma_1\sigma_2}(k_{1'} = k_1 + k_2 - k_{2'}, k_{2'}|k_1, k_2), \tag{A.15a}$$

$$G_{\sigma_2|\sigma_{2'}}(k_2|k_{2'}) = \delta_{k_2, k_{2'}} G_{\sigma_2|\sigma_{2'}}(k_2). \tag{A.15b}$$

Inserting Eqs. (A.15) into (A.14) yields

$$[A \cdot G]_{1'|1} = \sum_{k_2} A_{\sigma_{1'}\sigma_{2'}|\sigma_1\sigma_2}(k_1, k_2|k_1, k_2) G_{\sigma_2|\sigma_{2'}}(k_2). \tag{A.16}$$

In other words, we see that translational invariance and momentum conservation induce that the vertex $A$ inside the loop product $A \cdot G$ comes with $k_{1'} = k_1$ and $k_{2'} = k_2$. As can be understood from Fig. 7, the condition $k_{1'} = k_1$ imposes that $Q_{ph} = 0$, which makes the $ph$ convention particularly convenient to parametrize the momentum and frequency dependence of $A_{\sigma_{1'}\sigma_{2'}|\sigma_1\sigma_2}(k_1, k_2|k_1, k_2)$.

We thus set $A_{\sigma_{1'}\sigma_{2'}|\sigma_1\sigma_2}(k_1, k_2|k_1, k_2) = A_{\sigma_{1'}\sigma_{2'}|\sigma_1\sigma_2}(Q_{ph} = 0, k_{ph} = k_2, k'_{ph} = k_1)$, which enables us to rewrite the equation above as

$$[A \cdot G]_{1'|1} = \sum_{k_{ph}} A_{\sigma_{1'}\sigma_{2'}|\sigma_1\sigma_2}(Q_{ph} = 0, k_{ph}, k'_{ph}) G_{\sigma_2|\sigma_{2'}}(k_{ph}). \tag{A.17}$$

Alternatively, one can also use the crossing symmetry of $A$ to rewrite Eq. (A.14) as

$$[A \cdot G]_{1'|1} = A_{2'1'|21} G_{2|2'}. \tag{A.18}$$

This has the effect of exchanging the roles of $k_{ph}$ and $k'_{ph}$ in Eq. (A.17) and therefore yields

$$[A \cdot G]_{1'|1} = \sum_{k'_{ph}} A_{\sigma_{2'}\sigma_{1'}|\sigma_2\sigma_1}(Q_{ph} = 0, k_{ph}, k'_{ph}) G_{\sigma_2|\sigma_{2'}}(k'_{ph}). \tag{A.19}$$

We now turn to the second version of the loop product, Eq. (7), which plays a key role in the SDE formulation. Specifically, from Eq. (20), it is clear that in the *ph* channel formulation we consider

$$[G \cdot A]_{1'|1} = G_{2|2'} A_{2'1'|12} = \sum_{k_2,k_{2'}} G_{\sigma_2|\sigma_{2'}}(k_2|k_{2'}) A_{\sigma_{2'}\sigma_{1'}|\sigma_1\sigma_2}(k_{2'},k_{1'}|k_1,k_2). \tag{A.20}$$

Translational invariance and energy conservation implies

$$[A \cdot G]_{1'|1} = \sum_{k_2} G_{\sigma_2|\sigma_{2'}}(k_2) A_{\sigma_{2'}\sigma_{1'}|\sigma_1\sigma_2}(k_2,k_1|k_1,k_2). \tag{A.21}$$

As before, the loop product imposes that $k_{1'} = k_1$ and $k_{2'} = k_2$ for $A$. As a consequence, we conclude this time that the use of the $\overline{ph}$ convention to parametrize the momentum and frequency dependence of $A_{\sigma_{2'}\sigma_{1'}|\sigma_1\sigma_2}(k_2,k_1|k_1,k_2)$ is the most convenient choice, yielding

$$[A \cdot G]_{1'|1} = \sum_{k_{\overline{ph}}} G_{\sigma_2|\sigma_{2'}}(k_{\overline{ph}}) A_{\sigma_{2'}\sigma_{1'}|\sigma_1\sigma_2}(Q_{\overline{ph}}=0, k_{\overline{ph}}, k'_{\overline{ph}}). \tag{A.22}$$

As above, the crossing symmetry of $A$ allows us to write equivalently

$$[A \cdot G]_{1'|1} = \sum_{k'_{\overline{ph}}} G_{\sigma_2|\sigma_{2'}}(k'_{\overline{ph}}) A_{\sigma_{1'}\sigma_{2'}|\sigma_1\sigma_2}(Q_{\overline{ph}}=0, k_{\overline{ph}}, k'_{\overline{ph}}). \tag{A.23}$$

As a next step, we focus on relations that enable us to derive the flow equations for the bosonic propagators, the fermion-boson couplings and for the rest functions in Appendix D. In other words, we show that the following relations hold for the $\overline{ph}$ channel

$$\left[A \circ \Pi_{\overline{ph}}\right](Q_{\overline{ph}}, k_{\overline{ph}}, k'_{\overline{ph}}) = A(Q_{\overline{ph}}, k_{\overline{ph}}, k'_{\overline{ph}}) \bullet \Pi_{\overline{ph}}(Q_{\overline{ph}}, k'_{\overline{ph}}), \tag{A.24a}$$

$$\left[A \circ \Pi_{\overline{ph}} \circ B\right](Q_{\overline{ph}}, k_{\overline{ph}}, k'_{\overline{ph}}) = \sum_{k''_{\overline{ph}}} A(Q_{\overline{ph}}, k_{\overline{ph}}, k''_{\overline{ph}}) \bullet \Pi_{\overline{ph}}(Q_{\overline{ph}}, k''_{\overline{ph}}) \bullet B(Q_{\overline{ph}}, k''_{\overline{ph}}, k'_{\overline{ph}}), \tag{A.24b}$$

and similarly for the $\overline{ph}$ and $pp$ channel

$$\left[A \circ \Pi_{ph}\right](Q_{ph}, k_{ph}, k'_{ph}) = A(Q_{ph}, k_{ph}, k'_{ph}) \bullet \Pi_{ph}(Q_{ph}, k'_{ph}), \tag{A.25a}$$

$$\left[A \circ \Pi_{ph} \circ B\right](Q_{ph}, k_{ph}, k'_{ph}) = \sum_{k''_{ph}} A(Q_{ph}, k_{ph}, k''_{ph}) \bullet \Pi_{ph}(Q_{ph}, k''_{ph}) \bullet B(Q_{ph}, k''_{ph}, k'_{ph}), \tag{A.25b}$$

and respectively

$$\left[A \circ \Pi_{pp}\right](Q_{pp}, k_{pp}, k'_{pp}) = A(Q_{pp}, k_{pp}, k'_{pp}) \bullet \Pi_{pp}(Q_{pp}, k_{pp}), \tag{A.26a}$$

$$\left[A \circ \Pi_{pp} \circ B\right](Q_{pp}, k_{pp}, k'_{pp}) = \sum_{k''_{pp}} A(Q_{pp}, k''_{pp}, k'_{pp}) \bullet \Pi_{pp}(Q_{pp}, k''_{pp}) \bullet B(Q_{pp}, k_{pp}, k''_{pp}). \tag{A.26b}$$

Here, $A$ and $B$ are generic two-particle objects. For the derivation of these equations, we first consider products of the form

$$\left[A \circ \Pi_{\overline{ph}}\right]_{12|34} = A_{16|54}\Pi_{\overline{ph};52|36} = \sum_{k_5,k_6} A_{\sigma_1\sigma_6|\sigma_5\sigma_4}(k_1,k_6|k_5,k_4)\Pi_{\overline{ph};\sigma_5\sigma_2|\sigma_3\sigma_6}(k_5,k_2|k_3,k_6), \tag{A.27}$$

from Eq. (A.24a), where we restrict ourselves to the $\overline{ph}$ channel for simplicity. Using

$$A_{\sigma_1\sigma_6|\sigma_5\sigma_4}(k_1,k_6|k_5,k_4) = \delta_{k_1+k_6,k_5+k_4} A_{\sigma_1\sigma_6|\sigma_5\sigma_4}\left(Q_{\overline{ph}}, k_{\overline{ph}}, k^{(1)}_{\overline{ph}}\right), \tag{A.28}$$

and the channel-dependent bubble

$$\Pi_{\overline{ph};\sigma_5\sigma_2|\sigma_3\sigma_6}(k_5,k_2|k_3,k_6) = \delta_{k_5,k_3}\delta_{k_2,k_6}G_{\sigma_5|\sigma_3}(k_3)G_{\sigma_2|\sigma_6}(k_2)\,, \qquad (A.29)$$

we obtain

$$\left[A\circ\Pi_{\overline{ph}}\right]_{12|34} = \delta_{k_1+k_2,k_3+k_4}A_{\sigma_1\sigma_6|\sigma_5\sigma_4}\left(Q_{\overline{ph}},k_{\overline{ph}},k_{\overline{ph}}^{(1)}\right)G_{\sigma_5|\sigma_3}(k_3)G_{\sigma_2|\sigma_6}(k_2)\,, \qquad (A.30)$$

where the parametrization in terms of $Q_{\overline{ph}}$, $k_{\overline{ph}}$, and $k_{\overline{ph}}^{(1)}$ of the two-particle vertex follows the conventions specified in Fig. 7, with

$$\begin{cases} k_1 = \left(\mathbf{k}_{\overline{ph}}+\mathbf{Q}_{\overline{ph}},\, \nu_{\overline{ph}}+\left\lfloor\frac{\Omega_{\overline{ph}}}{2}\right\rfloor\right)\,, \\ k_6 = \left(\mathbf{k}_{\overline{ph}}^{(1)},\, \nu_{\overline{ph}}^{(1)}-\left\lceil\frac{\Omega_{\overline{ph}}}{2}\right\rceil\right)\,, \\ k_5 = \left(\mathbf{k}_{\overline{ph}}^{(1)}+\mathbf{Q}_{\overline{ph}},\, \nu_{\overline{ph}}^{(1)}+\left\lfloor\frac{\Omega_{\overline{ph}}}{2}\right\rfloor\right)\,, \\ k_4 = \left(\mathbf{k}_{\overline{ph}},\, \nu_{\overline{ph}}-\left\lceil\frac{\Omega_{\overline{ph}}}{2}\right\rceil\right)\,. \end{cases} \qquad (A.31)$$

At the same time, it holds

$$\begin{aligned} \left[A\circ\Pi_{\overline{ph}}\right]_{12|34} &= \left[A\circ\Pi_{\overline{ph}}\right]_{\sigma_1\sigma_2|\sigma_3\sigma_4}(k_1,k_2|k_3,k_4) \\ &= \delta_{k_1+k_2,k_3+k_4}\left[A\circ\Pi_{\overline{ph}}\right]_{\sigma_1\sigma_2|\sigma_3\sigma_4}\left(Q_{\overline{ph}}^{(1)},k_{\overline{ph}}^{(2)},k_{\overline{ph}}^{(3)}\right)\,, \end{aligned} \qquad (A.32)$$

where

$$\begin{cases} k_1 = \left(\mathbf{k}_{\overline{ph}}^{(2)}+\mathbf{Q}_{\overline{ph}}^{(1)},\, \nu_{\overline{ph}}^{(2)}+\left\lfloor\frac{\Omega_{\overline{ph}}^{(1)}}{2}\right\rfloor\right)\,, \\ k_2 = \left(\mathbf{k}_{\overline{ph}}^{(3)},\, \nu_{\overline{ph}}^{(3)}-\left\lceil\frac{\Omega_{\overline{ph}}^{(1)}}{2}\right\rceil\right)\,, \\ k_3 = \left(\mathbf{k}_{\overline{ph}}^{(3)}+\mathbf{Q}_{\overline{ph}}^{(1)},\, \nu_{\overline{ph}}^{(3)}+\left\lfloor\frac{\Omega_{\overline{ph}}^{(1)}}{2}\right\rfloor\right)\,, \\ k_4 = \left(\mathbf{k}_{\overline{ph}}^{(2)},\, \nu_{\overline{ph}}^{(2)}-\left\lceil\frac{\Omega_{\overline{ph}}^{(1)}}{2}\right\rceil\right)\,. \end{cases} \qquad (A.33)$$

We find

$$\left[A\circ\Pi_{\overline{ph}}\right]_{\sigma_1\sigma_2|\sigma_3\sigma_4}\left(Q_{\overline{ph}},k_{\overline{ph}},k_{\overline{ph}}^{(1)}\right) = A_{\sigma_1\sigma_6|\sigma_5\sigma_4}\left(Q_{\overline{ph}},k_{\overline{ph}},k_{\overline{ph}}^{(1)}\right)\Pi_{\overline{ph};\sigma_5\sigma_2|\sigma_3\sigma_6}\left(Q_{\overline{ph}},k_{\overline{ph}}^{(1)}\right)\,, \qquad (A.34)$$

as stated in Eq. (A.24a), where

$$\Pi_{\overline{ph};\sigma_1\sigma_2|\sigma_3\sigma_4}\left(Q_{\overline{ph}},k_{\overline{ph}}^{(1)}\right) = G_{\sigma_1|\sigma_3}\left(\mathbf{k}_{\overline{ph}}^{(1)}+\mathbf{Q}_{\overline{ph}},\, \nu_{\overline{ph}}^{(1)}+\left\lfloor\frac{\Omega_{\overline{ph}}}{2}\right\rfloor\right)G_{\sigma_2|\sigma_4}\left(\mathbf{k}_{\overline{ph}}^{(1)},\, \nu_{\overline{ph}}^{(1)}-\left\lceil\frac{\Omega_{\overline{ph}}}{2}\right\rceil\right)\,. \qquad (A.35)$$

Analogously, the above relation can be easily extended to the $\overline{ph}$ and $pp$ channel.

We also consider products involving an additional $B$ as in Eq. (A.24b). Starting from

$$\begin{aligned} \left[A\circ\Pi_{\overline{ph}}\circ B\right]_{12|34} &= \left[A\circ\Pi_{\overline{ph}}\right]_{16|54}B_{52|36} \\ &= \sum_{k_5,k_6}\left[A\circ\Pi_{\overline{ph}}\right]_{\sigma_1\sigma_6|\sigma_5\sigma_4}(k_1,k_6|k_5,k_4)B_{\sigma_5\sigma_2|\sigma_3\sigma_6}(k_5,k_2|k_3,k_6)\,, \end{aligned} \qquad (A.36)$$

we set

$$\begin{cases} k_1 = \left(\mathbf{k}_{\overline{ph}} + \mathbf{Q}_{\overline{ph}}, \nu_{\overline{ph}} + \left\lfloor \frac{\Omega_{\overline{ph}}}{2} \right\rfloor \right), \\ k_6 = \left(\mathbf{k}_{ph}^{(1)}, \nu_{ph}^{(1)} - \left\lceil \frac{\Omega_{\overline{ph}}}{2} \right\rceil \right), \\ k_5 = \left(\mathbf{k}_{ph}^{(1)} + \mathbf{Q}_{\overline{ph}}, \nu_{ph}^{(1)} + \left\lfloor \frac{\Omega_{\overline{ph}}}{2} \right\rfloor \right), \\ k_4 = \left(\mathbf{k}_{\overline{ph}}, \nu_{\overline{ph}} - \left\lceil \frac{\Omega_{\overline{ph}}}{2} \right\rceil \right), \end{cases} \qquad \begin{cases} k_5 = \left(\mathbf{k}_{ph}^{(2)} + \mathbf{Q}_{ph}^{(1)}, \nu_{ph}^{(2)} + \left\lfloor \frac{\Omega_{ph}^{(1)}}{2} \right\rfloor \right), \\ k_2 = \left(\mathbf{k}_{ph}^{(3)}, \nu_{ph}^{(3)} - \left\lceil \frac{\Omega_{ph}^{(1)}}{2} \right\rceil \right), \\ k_3 = \left(\mathbf{k}_{ph}^{(3)} + \mathbf{Q}_{ph}^{(1)}, \nu_{ph}^{(3)} + \left\lfloor \frac{\Omega_{ph}^{(1)}}{2} \right\rfloor \right), \\ k_6 = \left(\mathbf{k}_{ph}^{(2)}, \nu_{ph}^{(2)} - \left\lceil \frac{\Omega_{ph}^{(1)}}{2} \right\rceil \right). \end{cases} \tag{A.37}$$

With these specifications, we obtain

$$\left[A \circ \Pi_{\overline{ph}} \circ B\right]_{12|34} = \delta_{k_2 - k_3, k_1 - k_4} \sum_{k_5} \left[A \circ \Pi_{\overline{ph}}\right]_{\sigma_1 \sigma_6 | \sigma_5 \sigma_4} \left(Q_{\overline{ph}}, k_{\overline{ph}}, k_{ph}^{(1)}\right)$$
$$\times B_{\sigma_5 \sigma_2 | \sigma_3 \sigma_6} \left(Q_{\overline{ph}}, k_{ph}^{(1)}, k_{ph}^{(3)}\right). \tag{A.38}$$

In addition, we employ the relation

$$\left[A \circ \Pi_{\overline{ph}} \circ B\right]_{12|34} = \delta_{k_1 + k_2, k_3 + k_4} \left[A \circ \Pi_{\overline{ph}} \circ B\right]_{\sigma_1 \sigma_2 | \sigma_3 \sigma_4} \left(Q_{ph}^{(2)}, k_{ph}^{(4)}, k_{ph}^{(5)}\right), \tag{A.39}$$

with

$$\begin{cases} k_1 = \left(\mathbf{k}_{ph}^{(4)} + \mathbf{Q}_{ph}^{(2)}, \nu_{ph}^{(4)} + \left\lfloor \frac{\Omega_{ph}^{(2)}}{2} \right\rfloor \right), \\ k_2 = \left(\mathbf{k}_{ph}^{(5)}, \nu_{ph}^{(5)} - \left\lceil \frac{\Omega_{ph}^{(2)}}{2} \right\rceil \right), \\ k_3 = \left(\mathbf{k}_{ph}^{(5)} + \mathbf{Q}_{ph}^{(2)}, \nu_{ph}^{(5)} + \left\lfloor \frac{\Omega_{ph}^{(2)}}{2} \right\rfloor \right), \\ k_4 = \left(\mathbf{k}_{ph}^{(4)}, \nu_{ph}^{(4)} - \left\lceil \frac{\Omega_{ph}^{(2)}}{2} \right\rceil \right). \end{cases} \tag{A.40}$$

Thus, we infer

$$\left[A \circ \Pi_{\overline{ph}} \circ B\right]_{12|34} = \delta_{k_1 + k_2, k_3 + k_4} \left[A \circ \Pi_{\overline{ph}} \circ B\right]_{\sigma_1 \sigma_2 | \sigma_3 \sigma_4} \left(Q_{\overline{ph}}, k_{\overline{ph}}, k_{ph}^{(3)}\right). \tag{A.41}$$

Comparing Eqs. (A.38) and (A.41) yields the anticipated result, i.e., Eq. (A.24b). This is evident by relabelling $k_{\overline{ph}}^{(1)}$ by $k_{\overline{ph}}^{(2)}$ and $k_{\overline{ph}}^{(3)}$ by $k_{ph}^{(1)}$

$$\left[A \circ \Pi_{\overline{ph}} \circ B\right]_{\sigma_1 \sigma_2 | \sigma_3 \sigma_4} \left(Q_{\overline{ph}}, k_{\overline{ph}}, k_{ph}^{(1)}\right) = \sum_{k_{\overline{ph}}^{(2)}} \left[A \circ \Pi_{\overline{ph}}\right]_{\sigma_1 \sigma_6 | \sigma_5 \sigma_4} \left(Q_{\overline{ph}}, k_{\overline{ph}}, k_{ph}^{(2)}\right)$$
$$\times B_{\sigma_1 \sigma_2 | \sigma_3 \sigma_6} \left(Q, k_{ph}^{(2)}, k_{ph}^{(1)}\right). \tag{A.42}$$

# B   Extension to non-local interactions

In the presence of non-local bare interactions of the generic form $U = U(Q_r, k_r, k_r')$, a naive application of the single-boson exchange decomposition based on the classification of diagrams

in terms of $U$ reducibility yields bosonic propagators $w_r(Q_r, k_r, k_r')$ and fermion-boson couplings $\lambda_r(Q_r, k_r, k_r')$ with the full momentum and frequency dependence, spoiling its original idea.

For the extended Hubbard model with an additional nearest-neighbor interaction, this can be overcome by considering a generalized single-boson exchange formulation [57], where the notion of bare interaction reducibility is replaced by a $\mathcal{B}$ reducibility: the bare interaction in each channel is split according to

$$U_r(Q_r, k_r, k_r') = \mathcal{B}_r(Q_r) + \mathcal{F}_r(Q_r, k_r, k_r'), \tag{B.1}$$

where $\mathcal{B}_r$ depends exclusively on the bosonic momentum and frequency in channel $r$, while $\mathcal{F}_r$ carries the dependence on the fermionic arguments. The bosonic propagator $w_r^{(\mathcal{B})}(Q_r)$ and the fermion-boson coupling $\lambda_r^{(\mathcal{B})}(Q_r, k_r)$ then retain their reduced momentum and frequency dependence characteristic of the single-boson exchange formulation[4] (we here introduced an additional superscript to disambiguate them from the $w_r$ and $\lambda_r$ for local interactions referred to in the main text). However, the relation (14) does not hold anymore in this case

$$w_r^{(\mathcal{B})} \bullet \lambda_r^{(\mathcal{B})} \neq U_r + U_r \circ \Pi_r \circ V_r. \tag{B.2}$$

For the generalized single-boson exchange formulation we have instead

$$w_r^{(\mathcal{B})} \bullet \lambda_r^{(\mathcal{B})} = \mathcal{B}_r + \mathcal{B}_r \circ \Pi_r \circ V_r. \tag{B.3}$$

As a consequence, the SDE will not reduce to the form derived for local interactions. In particular, inserting Eq. (B.1) in the conventional form of the SDE leads to an additional term of the form $\mathcal{F}_r \circ \Pi_r \circ V_r$ that cannot be absorbed in a product of $w_r^{(\mathcal{B})}$ and $\lambda_r^{(\mathcal{B})}$. However, if $\mathcal{F}_r \circ \Pi_r \circ V_r = 0$, the results of the main text still apply. In fact, this applies for the extended Hubbard model in the $s$-wave truncation [57].

# C  Momentum and frequency dependence of the SDE

We here outline the derivation of the SDE in the form of Eqs. (25) with the explicit momentum and frequency dependence. Starting from Eqs. (21) derived in the main text, Eq. (A.17) introduced in Appendix A.2 allows us to rewrite the SDE as

$$\Sigma_{\sigma_{1'}|\sigma_1}(k_{ph}') = \sum_{k_{ph}} A_{\sigma_{1'}\sigma_{2'}|\sigma_1\sigma_2}(Q_{ph} = 0, k_{ph}, k_{ph}')G_{\sigma_2|\sigma_{2'}}(k_{ph}). \tag{C.1}$$

Specifically, we focus on the $\overline{ph}$ channel formulation first. In order to directly compare to Eq. (21a), where the spin components for the various terms contributing to the SDE are already fixed, we rewrite Eq. (C.1) for $\sigma_{1'} = \sigma_1 = \uparrow$ and $\sigma_{2'} = \sigma_2 = \downarrow$, namely

$$\Sigma^{\uparrow}(k_{ph}') = \sum_{k_{ph}} A_{\uparrow\downarrow}(Q_{ph} = 0, k_{ph}, k_{ph}')G^{\downarrow}(k_{ph}), \tag{C.2}$$

where we also used the shorthand notation (16) to express $\Sigma^{\uparrow|\uparrow} = \Sigma^{\uparrow}$, $A_{\uparrow\downarrow|\uparrow\downarrow} = A_{\uparrow\downarrow}$, and $G_{\downarrow|\downarrow} = G^{\downarrow}$. We stress that Eq. (C.2) does not involve any summation over spin indices, like Eqs. (21). We can thus write Eq. (21a) in the form (C.2) by setting

$$A_{\uparrow\downarrow}(Q_{ph} = 0, k_{ph}, k_{ph}') = \left[ -w_{\overline{ph}}^{\uparrow\downarrow}(Q_{\overline{ph}})\lambda_{\overline{ph}}^{\uparrow\downarrow}(Q_{\overline{ph}}, k_{\overline{ph}}') \right](Q_{ph} = 0, k_{ph}, k_{ph}'). \tag{C.3}$$

---

[4]Note that in Ref. [9], $w_r$ and $\lambda_r$ are defined by separating the $U$-reducible parts of the full vertex $V$, regardless of the momentum dependence of $U$. In contrast, $w_r^{(\mathcal{B})}$ and $\lambda_r^{(\mathcal{B})}$ are defined with respect to $\mathcal{B}$ reducibility with a reduced momentum and frequency dependence. In this sense they represent a generalization of $w_r$ and $\lambda_r$ for non-local interactions.

In order to determine the momentum and frequency dependence of $w_{\underline{ph}}^{\uparrow\downarrow}$ and $\lambda_{\underline{ph}}^{\uparrow\downarrow}$, we translate $Q_{\overline{ph}}$ and $k'_{\overline{ph}}$ into the $ph$ notation with $Q_{ph} = 0$ by using the following relations from Fig. 7

$$Q_{\overline{ph}} = (\mathbf{Q}_{\overline{ph}}, \Omega_{\overline{ph}})\underset{Q_{ph}=0}{=} (\mathbf{k}'_{ph} - \mathbf{k}_{ph}, \nu'_{ph} - \nu_{ph}), \tag{C.4a}$$

$$k'_{\overline{ph}} = (\mathbf{k}'_{\overline{ph}}, \nu'_{\overline{ph}})\underset{Q_{ph}=0}{=} \left(\mathbf{k}_{ph}, \left\lceil \frac{\nu_{ph} + \nu'_{ph}}{2} \right\rceil_{\text{ferm}}\right), \tag{C.4b}$$

where we introduced the notation $\lceil ... \rceil_{\text{ferm}}$ $\left(\lfloor ... \rfloor_{\text{ferm}}\right)$ which rounds its argument up (down) to the nearest *fermionic* Matsubara frequency. This differs from the symbols $\lceil ... \rceil$ and $\lfloor ... \rfloor$ used previously to round up or down to the nearest *bosonic* Matsubara frequency. For clarity, these symbols will be replaced by $\lceil ... \rceil_{\text{bos}}$ and $\lfloor ... \rfloor_{\text{bos}}$ respectively in the following. Hence, we can rewrite Eq. (C.3) as

$$A_{\uparrow\downarrow}(Q_{ph} = 0, k_{ph}, k'_{ph}) = -w_{\underline{ph}}^{\uparrow\downarrow}(\mathbf{k}'_{ph} - \mathbf{k}_{ph}; \nu'_{ph} - \nu_{ph})$$
$$\times \lambda_{\underline{ph}}^{\uparrow\downarrow}\left(\mathbf{k}'_{ph} - \mathbf{k}_{ph}, \mathbf{k}_{ph}; \nu'_{ph} - \nu_{ph}, \left\lceil \frac{\nu_{ph} + \nu'_{ph}}{2} \right\rceil_{\text{ferm}}\right). \tag{C.5}$$

The self-energy, Eq. (C.2), then reads

$$\Sigma^{\uparrow}(\mathbf{k}; \nu) = -\sum_{\mathbf{k}', \nu'} w_{\underline{ph}}^{\uparrow\downarrow}(\mathbf{k} - \mathbf{k}'; \nu - \nu')\lambda_{\underline{ph}}^{\uparrow\downarrow}\left(\mathbf{k} - \mathbf{k}', \mathbf{k}'; \nu - \nu', \left\lceil \frac{\nu + \nu'}{2} \right\rceil_{\text{ferm}}\right)G^{\downarrow}(\mathbf{k}'; \nu'), \tag{C.6}$$

where the momentum and frequency indices have been relabeled. Setting $\mathbf{Q} = \mathbf{k} - \mathbf{k}'$ and $\Omega = \nu - \nu'$, the fermionic frequency argument of $\lambda^{M}$ can be expressed as

$$\left\lceil \frac{\nu + \nu'}{2} \right\rceil_{\text{ferm}} = \left\lceil \frac{2\nu - \Omega}{2} \right\rceil_{\text{ferm}} = \nu - \left\lceil \frac{\Omega}{2} \right\rceil_{\text{bos}}. \tag{C.7}$$

With this, the right-hand side of Eq. (C.6) can be rewritten as a sum over the bosonic arguments $\mathbf{Q}$ and $\Omega$ by

$$\Sigma^{\uparrow}(\mathbf{k}; \nu) = -\sum_{\mathbf{Q}, \Omega} w_{\underline{ph}}^{\uparrow\downarrow}(\mathbf{Q}; \Omega)\lambda_{\underline{ph}}^{\uparrow\downarrow}\left(\mathbf{Q}, \mathbf{k} - \mathbf{Q}; \Omega, \nu - \left\lceil \frac{\Omega}{2} \right\rceil_{\text{bos}}\right)G^{\downarrow}(\mathbf{k} - \mathbf{Q}; \nu - \Omega). \tag{C.8}$$

As explained in Appendix A.2, one can also use crossing symmetry to obtain Eq. (A.19), for which the starting point of our derivation is

$$\Sigma^{\uparrow}(k_{ph}) = \sum_{k'_{ph}} A_{\uparrow\downarrow}(Q_{ph} = 0, k_{ph}, k'_{ph})G^{\downarrow}(k'_{ph}), \tag{C.9}$$

instead of Eq. (C.2). This modifies the arguments in Eq. (C.8) which are substituted by

$$\Sigma^{\uparrow}(\mathbf{k}; \nu) = -\sum_{\mathbf{Q}, \Omega} w_{\underline{ph}}^{\uparrow\downarrow}(\mathbf{Q}; \Omega)\lambda_{\underline{ph}}^{\uparrow\downarrow}\left(\mathbf{Q}, \mathbf{k}; \Omega, \nu + \left\lceil \frac{\Omega}{2} \right\rceil_{\text{bos}}\right)G^{\downarrow}(\mathbf{k} + \mathbf{Q}; \nu + \Omega). \tag{C.10}$$

We note that Eqs. (C.8) and (C.10) are fully equivalent since they are only related by crossing symmetry.

A similar result can be derived for the $ph$ channel formulation by starting from Eqs. (20) and (A.22). From their comparison we infer

$$\Sigma^{\uparrow}(k'_{\overline{ph}}) = \sum_{k_{\overline{ph}}} G^{\downarrow}(k_{\overline{ph}})A_{\widehat{\downarrow\uparrow}}(Q_{\overline{ph}} = 0, k_{\overline{ph}}, k'_{\overline{ph}}). \tag{C.11}$$

Thus, we identify

$$A_{\widehat{\downarrow\uparrow}}(Q_{\overline{ph}}=0,k_{\overline{ph}},k'_{\overline{ph}}) = \left[w^{\widehat{\downarrow\uparrow}}_{ph}(Q_{ph})\lambda^{\widehat{\downarrow\uparrow}}_{ph}(Q_{ph},k'_{ph})\right](Q_{\overline{ph}}=0,k_{\overline{ph}},k'_{\overline{ph}}).\tag{C.12}$$

Following the same steps as above, and applying crossing symmetry, we obtain two different, yet equivalent expressions in the *ph* channel

$$\Sigma^{\uparrow}(\mathbf{k};\nu) = \sum_{\mathbf{Q},\Omega} w^{\widehat{\downarrow\uparrow}}_{ph}(\mathbf{Q};\Omega)\lambda^{\widehat{\downarrow\uparrow}}_{ph}\left(\mathbf{Q},\mathbf{k}-\mathbf{Q};\Omega,\nu-\left\lceil\frac{\Omega}{2}\right\rceil_{\mathrm{bos}}\right)G^{\downarrow}(\mathbf{k}-\mathbf{Q};\nu-\Omega),\tag{C.13a}$$

$$\Sigma^{\uparrow}(\mathbf{k};\nu) = \sum_{\mathbf{Q},\Omega} w^{\widehat{\downarrow\uparrow}}_{ph}(\mathbf{Q};\Omega)\lambda^{\widehat{\downarrow\uparrow}}_{ph}\left(\mathbf{Q},\mathbf{k};\Omega,\nu+\left\lceil\frac{\Omega}{2}\right\rceil_{\mathrm{bos}}\right)G^{\downarrow}(\mathbf{k}+\mathbf{Q};\nu+\Omega).\tag{C.13b}$$

The derivation in the superconducting channel is similar. In this case, we use the translation from the *pp* to the *ph* notation. Alternatively, it is also possible to start from Eq. (C.2), with

$$A_{\uparrow\downarrow}(Q_{ph}=0,k_{ph},k'_{ph}) = \left[-w^{\uparrow\downarrow}_{pp}(Q_{pp})\left(2\lambda^{\uparrow\downarrow}_{pp}(Q_{pp},k'_{pp})-1\right)\right](Q_{ph}=0,k_{ph},k'_{ph}).\tag{C.14}$$

From Fig. 7 we infer

$$Q_{pp} = (\mathbf{Q}_{pp},\Omega_{pp})\underset{Q_{ph}=0}{=}(\mathbf{k}_{ph}+\mathbf{k}'_{ph},\nu_{ph}+\nu'_{ph}),\tag{C.15a}$$

$$k'_{pp} = (\mathbf{k}'_{pp},\nu'_{pp})\underset{Q_{ph}=0}{=}\left(\mathbf{k}_{ph},\left\lceil\frac{\nu_{ph}-\nu'_{ph}}{2}\right\rceil_{\mathrm{ferm}}\right),\tag{C.15b}$$

leading to

$$\Sigma^{\uparrow}(\mathbf{k};\nu) = -\sum_{\mathbf{k}',\nu'} w^{\uparrow\downarrow}_{pp}(\mathbf{k}+\mathbf{k}';\nu+\nu')\left(2\lambda^{\uparrow\downarrow}_{pp}\left(\mathbf{k}+\mathbf{k}',\mathbf{k}';\nu+\nu',\left\lceil\frac{\nu'-\nu}{2}\right\rceil_{\mathrm{ferm}}\right)-1\right)G^{\downarrow}(\mathbf{k}';\nu').\tag{C.16}$$

By introducing the bosonic arguments $\mathbf{Q}=\mathbf{k}+\mathbf{k}'$ and $\Omega=\nu+\nu'$, this can be rewritten as

$$\Sigma^{\uparrow}(\mathbf{k};\nu) = -\sum_{\mathbf{Q},\Omega} w^{\uparrow\downarrow}_{pp}(\mathbf{Q};\Omega)\left(2\lambda^{\uparrow\downarrow}_{pp}\left(\mathbf{Q},\mathbf{Q}-\mathbf{k};\Omega,\left\lceil\frac{\Omega}{2}\right\rceil_{\mathrm{bos}}-\nu\right)-1\right)G^{\downarrow}(\mathbf{Q}-\mathbf{k};\Omega-\nu).\tag{C.17}$$

As before, crossing symmetry can be used to determine the equivalent expression

$$\Sigma^{\uparrow}(\mathbf{k};\nu) = -\sum_{\mathbf{Q},\Omega} w^{\uparrow\downarrow}_{pp}(\mathbf{Q};\Omega)\left(2\lambda^{\uparrow\downarrow}_{pp}\left(\mathbf{Q},\mathbf{k};\Omega,\nu-\left\lceil\frac{\Omega}{2}\right\rceil_{\mathrm{bos}}\right)-1\right)G^{\downarrow}(\mathbf{Q}-\mathbf{k};\Omega-\nu).\tag{C.18}$$

We now outline the derivation within the physical channel formulations, as presented in Eq. (36) in the main text, showing how it follows straightforwardly from the above lines assuming SU(2) symmetry. First, we focus on the magnetic channel formulation. In Eq. (C.3), we can identify

$$A_{\uparrow\downarrow} = -w^{\uparrow\downarrow}_{ph}(Q_{\overline{ph}})\lambda^{\uparrow\downarrow}_{ph}(Q_{\overline{ph}},k'_{\overline{ph}}) = w^{\mathrm{M}}(Q_{\overline{ph}})\lambda^{\mathrm{M}}(Q_{\overline{ph}},k'_{\overline{ph}}).\tag{C.19}$$

Thus, following the same steps that led us to recover the final forms in Eqs. (C.8) and (C.10), we can derive the two equivalent magnetic channel formulations, which are related by crossing symmetry. With the explicit momentum and frequency dependencies, they read

$$\Sigma(\mathbf{k};\nu) = \sum_{\mathbf{Q},\Omega} w^{\mathrm{M}}(\mathbf{Q};\Omega)\lambda^{\mathrm{M}}\left(\mathbf{Q},\mathbf{k}-\mathbf{Q};\Omega,\nu-\left\lceil\frac{\Omega}{2}\right\rceil_{\mathrm{bos}}\right)G(\mathbf{k}-\mathbf{Q};\nu-\Omega)\tag{C.20a}$$

$$= \sum_{\mathbf{Q},\Omega} w^{\mathrm{M}}(\mathbf{Q};\Omega)\lambda^{\mathrm{M}}\left(\mathbf{Q},\mathbf{k};\Omega,\nu+\left\lceil\frac{\Omega}{2}\right\rceil_{\mathrm{bos}}\right)G(\mathbf{k}+\mathbf{Q};\nu+\Omega).\tag{C.20b}$$

The same reasoning applies for the density channel formulation for which we also use the translation from the $\overline{ph}$ to the $ph$ parametrization to obtain the explicit form of the self-energy. Similarly, crossing symmetry yields two different, but equivalent expressions

$$\Sigma(\mathbf{k}; \nu) = \sum_{\mathbf{Q},\Omega}\left[w^{\mathrm{D}}(\mathbf{Q};\Omega)\lambda^{\mathrm{D}}\left(\mathbf{Q},\mathbf{k}-\mathbf{Q};\Omega,\nu-\left\lceil\frac{\Omega}{2}\right\rceil_{\mathrm{bos}}\right) - 2U^{\mathrm{D}}(\mathbf{Q},\mathbf{k};\Omega,\nu)\right]G(\mathbf{k}-\mathbf{Q};\nu-\Omega), \quad \text{(C.21a)}$$

$$\Sigma(\mathbf{k}; \nu) = \sum_{\mathbf{Q},\Omega}\left[w^{\mathrm{D}}(\mathbf{Q};\Omega)\lambda^{\mathrm{D}}\left(\mathbf{Q},\mathbf{k};\Omega,\nu+\left\lceil\frac{\Omega}{2}\right\rceil_{\mathrm{bos}}\right) - 2U^{\mathrm{D}}(\mathbf{Q},\mathbf{k};\Omega,\nu)\right]G(\mathbf{k}+\mathbf{Q};\nu+\Omega). \quad \text{(C.21b)}$$

For the superconducting channel, the key point is the identification of

$$A_{\uparrow\downarrow} = -w_{pp}^{\uparrow\downarrow}(Q_{pp})\left(2\lambda_{pp}^{\uparrow\downarrow}(Q_{pp},k'_{pp})-1\right) = -w^{\mathrm{SC}}(Q_{pp})\lambda^{\mathrm{SC}}(Q_{pp},k'_{pp}) \quad \text{(C.22)}$$

in Eq. (C.14). Analogously to the derivation of Eqs.(C.17) and (C.18), we obtain the two crossing symmetry-related equivalent formulations

$$\Sigma(\mathbf{k}; \nu) = -\sum_{\mathbf{Q},\Omega}w^{\mathrm{SC}}(\mathbf{Q};\Omega)\lambda^{\mathrm{SC}}\left(\mathbf{Q},\mathbf{Q}-\mathbf{k};\Omega,\left\lceil\frac{\Omega}{2}\right\rceil_{\mathrm{bos}}-\nu\right)G(\mathbf{Q}-\mathbf{k};\Omega-\nu) \quad \text{(C.23a)}$$

$$= -\sum_{\mathbf{Q},\Omega}w^{\mathrm{SC}}(\mathbf{Q};\Omega)\lambda^{\mathrm{SC}}\left(\mathbf{Q},\mathbf{k};\Omega,\nu-\left\lceil\frac{\Omega}{2}\right\rceil_{\mathrm{bos}}\right)G(\mathbf{Q}-\mathbf{k};\Omega-\nu). \quad \text{(C.23b)}$$

# D  Single-boson exchange flow equations

In this section, we report the $(1\ell)$ fRG equations for the bosonic propagators, the fermion-boson couplings, and the rest functions (the flow equation for the self-energy is obtained from the derivative of the SDE). In diagrammatic channels [9, 10, 15], they read

$$\dot{w}_r = w_r \bullet \lambda_r \circ \dot{\Pi}_r \circ \lambda_r \bullet w_r, \quad \text{(D.1a)}$$

$$\dot{\lambda}_r = \lambda_r \circ \dot{\Pi}_r \circ \mathcal{I}_r, \quad \text{(D.1b)}$$

$$\dot{M}_r = \mathcal{I}_r \circ \dot{\Pi}_r \circ \mathcal{I}_r, \quad \text{(D.1c)}$$

where $\mathcal{I}_r$ is the $U$ irreducible vertex in channel $r$.

In physical channels, the explicit form for the magnetic channel is[5]

$$\dot{w}^{\mathrm{M}}(Q) = -\dot{w}_{\overline{ph}}^{\uparrow\downarrow}(Q) = -\left[w^{\mathrm{M}}(Q)\right]^2\sum_k\lambda^{\mathrm{M}}(Q,k)\dot{\Pi}^{\mathrm{M}}(Q,k)\lambda^{\mathrm{M}}(Q,k), \quad \text{(D.2a)}$$

$$\dot{\lambda}^{\mathrm{M}}(Q,k) = \dot{\lambda}_{\overline{ph}}^{\uparrow\downarrow} = -\sum_{k'}\lambda^{\mathrm{M}}(Q,k')\dot{\Pi}^{\mathrm{M}}(Q,k')\mathcal{I}^{\mathrm{M}}(Q,k',k), \quad \text{(D.2b)}$$

$$\dot{M}^{\mathrm{M}}(Q,k,k') = -\dot{M}_{\overline{ph}}^{\uparrow\downarrow}(Q,k,k') = -\sum_{k''}\mathcal{I}^{\mathrm{M}}(Q,k,k'')\dot{\Pi}^{\mathrm{M}}(Q,k'')\mathcal{I}^{\mathrm{M}}(Q,k'',k'). \quad \text{(D.2c)}$$

Analogously, for the density channel we have

$$\dot{w}^{\mathrm{D}}(Q) = \dot{w}_{ph}^{\uparrow\uparrow}(Q) + \dot{w}_{ph}^{\uparrow\downarrow}(Q) = \left[w^{\mathrm{D}}(Q)\right]^2\sum_k\lambda^{\mathrm{D}}(Q,k)\dot{\Pi}^{\mathrm{D}}(Q,k)\lambda^{\mathrm{D}}(Q,k), \quad \text{(D.3a)}$$

$$\dot{\lambda}^{\mathrm{D}}(Q,k) = \dot{\lambda}_{ph}^{\uparrow\uparrow}(Q,k) + \dot{\lambda}_{ph}^{\uparrow\downarrow}(Q,k) = \sum_{k'}\lambda^{\mathrm{D}}(Q,k')\dot{\Pi}^{\mathrm{D}}(Q,k')\mathcal{I}^{\mathrm{D}}(Q,k',k), \quad \text{(D.3b)}$$

$$\dot{M}^{\mathrm{D}}(Q,k,k') = \dot{M}_{ph}^{\uparrow\uparrow}(Q,k,k') + \dot{M}_{ph}^{\uparrow\downarrow}(Q,k,k') = \sum_{k''}\mathcal{I}^{\mathrm{D}}(Q,k,k'')\dot{\Pi}^{\mathrm{D}}(Q,k'')\mathcal{I}^{\mathrm{D}}(Q,k'',k'), \quad \text{(D.3c)}$$

---

[5]For completeness, we here report also the flow equation for the rest function, despite it is neglected in the numerical results discussed in Section 3.

and for the superconducting channel

$$\dot{w}^{\text{SC}}(Q) = \dot{w}_{pp}^{\uparrow\downarrow}(Q) - \dot{w}_{pp}^{\widehat{\uparrow\downarrow}}(Q) = \left[ w^{\text{SC}}(Q) \right]^2 \sum_k \lambda^{\text{SC}}(Q,k) \dot{\Pi}^{\text{SC}}(Q,k) \lambda^{\text{SC}}(Q,k), \qquad \text{(D.4a)}$$

$$\dot{\lambda}^{\text{SC}}(Q,k) = \dot{\lambda}_{pp}^{\uparrow\downarrow}(Q,k) - \dot{\lambda}_{pp}^{\widehat{\uparrow\downarrow}}(Q,k) = \sum_{k'} \lambda^{\text{SC}}(Q,k') \dot{\Pi}^{\text{SC}}(Q,k') \mathcal{I}^{\text{SC}}(Q,k,k'), \qquad \text{(D.4b)}$$

$$\dot{M}^{\text{SC}}(Q,k,k') = \dot{M}_{pp}^{\uparrow\downarrow}(Q,k,k') - \dot{M}_{pp}^{\widehat{\uparrow\downarrow}}(Q,k,k') = \sum_{k''} \mathcal{I}^{\text{SC}}(Q,k'',k') \dot{\Pi}^{\text{SC}}(Q,k'') \mathcal{I}^{\text{SC}}(Q,k,k''),$$

$$\text{(D.4c)}$$

where we used the corresponding definitions in the physical channels for the bubbles, Eq. (3), as well as the considerations provided in Appendix A.

As an example, we illustrate how the flow equation for the bosonic propagator in the magnetic channel is obtained from Eq. (D.1a) through the use of Eq. (6b)

$$\begin{aligned}
\dot{w}^{\text{M}} = -\dot{w}_{\overline{ph}}^{\uparrow\downarrow} &= -\left[ w_{\overline{ph}} \bullet \lambda_{\overline{ph}} \circ \dot{\Pi}_{\overline{ph}} \circ \lambda_{\overline{ph}} \bullet w_{\overline{ph}} \right]^{\uparrow\downarrow} \\
&= -\left[ w_{\overline{ph}}^{\uparrow\downarrow} \right]^2 \left[ \lambda_{\overline{ph}} \circ \dot{\Pi}_{\overline{ph}} \circ \lambda_{\overline{ph}} \right]^{\uparrow\downarrow} \\
&= -\left[ w_{\overline{ph}}^{\uparrow\downarrow} \right]^2 \lambda_{\overline{ph}}^{\uparrow\downarrow} \circ \dot{\Pi}_{\overline{ph}}^{\uparrow\downarrow} \circ \lambda_{\overline{ph}}^{\uparrow\downarrow}.
\end{aligned} \qquad \text{(D.5)}$$

Up to now we focused on the spin structure, the momentum and frequency dependence as well as the respective summations still have to be considered. While for the ∘ product we can use Eq. (A.24b), the • multiplication with the bosonic propagator involves only the summation over spin indices. With the translation to the magnetic channel, Eq. (27), we thus recover Eq. (D.2a).

The derivation of the flow equations for the fermion-boson coupling (D.2b) and the rest function (D.2c) is straightforward, since these correspond to the ↑↓ spin component of the products in Eqs. (D.1b) and (D.1c) which are diagonal in the $\overline{ph}$ channel. The flow equations in the density and superconducting channels are obtained along the same lines. This applies also to the derivation of the momentum and frequency dependence of the multiloop equations, i.e., where Eqs. (D.1) are replaced by Eqs. (48) of Ref. [9].

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
