# Peer review of "Single-boson exchange formulation of the Schwinger-Dyson equation and its application to the functional renormalization group"

_SciPost Physics, doi:SciPost Phys. 18, 078 (2025)_

## Round 1 · Referee Report · Anonymous (Referee 1) · 2025-1-7

Strengths

See report

Weaknesses

See report

Report

The authors discuss the single-boson exchange formulation of the Schwinger-Dyson equation (SDE) and its application to the renormalization functional group (FRG). This paper is part of a long-standing effort to improve the fermionic FRG approach to the Hubbard model by explicitly including bosonic propagators in the theory (but without including bosonic fields in the functional integral). This type of approach is certainly very interesting and the article makes a useful contribution. Before publication, the authors should consider the following remarks.

1) In Eq.(1), the minus sign in front of the interaction term is unusual. For this sign to be correct, the interaction matrix element $U_{1'2'|12}$ must correspond to scattering $1\to 2'$ and $2\to 1'$ for the direct process (see, e.g. Negele-Orland Eq.(1.107a)). Is that correct?

2) The Hartree-Fock contribution $UG$ to the self-energy changes sign between (5) and (10).

3) It would be useful to give an example of diagrams that are two-particle reducible with respect to the channel $r$ and cannot be put in the form (13).

4) Is the Hartree-Fock term still included in Eq.(20)?

5) It would be useful to briefly describe the TU solvers.

6) It would also be useful to briefly recall the idea of the form-factor expansion in Sec.2.5.

7) The minus sign in (38) is unusual. This may be related to question (1).

8) I do not understand what is done in Sec.3. Are there self-energy corrections taken into account in the flow or is the self-energy computed after the flow? Also mixing FRG flow with SDE to compute the self-energy raises the following question. In the FRG, the energy scales are integrated out progressively so that the physics at scale $\Lambda$ depends only on the physics at scale $\Lambda+|d\Lambda|$. Is this property still true when one uses the SDE to compute the self-energy?

9) How do we see the pseudogap (or its absence) in Figs.3 and 4 showing the imaginary part of the Matsubara-frequency self-energy? Is it not necessary to consider the retarded self-energy to obtain information about the pseudogap?

10) In the conclusion, the authors could discuss the advantages and limitations of their approach with respect to the more traditional patch approach. One of the strengths of the latter is that it is unbiased since it treats all channels on equal footing. By contrast, it seems that there is some arbitrariness when using the single-boson exchange formulation of the Schwinger-Dyson equation. The problem is alluded to in Sec.3 and I understand that using the parquet-based formulation of the SDE might be the solution, but for the non-expert the statement is not very clear.

11) It would be helpful to illustrate Eq.(88) with an example, e.g. by considering a next-neighbor interaction.

Requested changes

See suggestions mentioned in the report

Recommendation

Ask for minor revision

  • validity: high
  • significance: good
  • originality: good
  • clarity: good
  • formatting: good
  • grammar: excellent

Author:  Miriam Patricolo  on 2025-02-04  [id 5186]

(in reply to Report 1 on 2025-01-07)

We thank the Referee for his assessment and valuable suggestions that helped to improve the manuscript.

In the following, we address the points raised by the Referee:

1) We use the minus sign in Eq. (1) in accordance to Refs. [9,48,54]. You are right that the interaction matrix element U1′2′|12 describes direct (particle-hole) scattering processes as 1 → 2′ and 2 → 1′. We have applied this convention consistently.

2) Concerning the change of sign in the Hartree-Fock contribution between Eqs. (5) and (10), we added a comment in the text, which explains its origin in terms of the crossing symmetry encoded in the formalism. We also rewrote the definition of the loop product G · A = −A · G in Eq. (7) to make the sign change between Eqs. (5) and (10) less confusing. Indeed, the sign change results from the crossing symmetry of operator A by exchanging two fermionic legs.

3) For the example of diagrams that are two-particle reducible with respect to the channel r that cannot be put in the form (13), we now explicitly refer to Fig. 1 of Ref. [10] and Fig. 5 in Ref. [9] illustrating this point.

4) The Hartree–Fock term is included in Eq. (20), the result is general. We added a sentence below it to clarify this point.

5-6) As suggested, we added a short description of the approximations used in TU solvers, together with the form-factor expansion in Sec. 2.5.

7) Exactly, in the convention we use (see point 1), we have U ↑↓ = −U .

8) Concerning the fRG computations of Sec. 3, we specified how the self- energy flow in the SDE formulation is implemented in the SDE formulation. This should clarify also how the successive integrating out of degrees of freedom can be correctly taken into account by the SDE form of the self-energy.

9) The pseudogap is typically associated to a feature in the spectral function, described by the retarded self-energy. We here refrain from performing ananalytic continuation and restrict ourselves to analyzing the fRG data obtained in Matsubara frequency space. Specifically, the presence of quasiparticles in a Fermi liquid is equivalent to a nonzero quasiparticle weight

                                                   Z(k) = [     1  −  ∂ReΣ(ν, k)/ ∂ν| ν→0     ]^( −1),

where ν is a real frequency. A non-Fermi-liquid behavior can be signaled by deviations from a nonzero Z(k) < 1, e.g. Z(k) → 0 or Z(k) > 1. In the low temperature limit, ∂ν ReΣ(ν, k)|ν→0 can be translated to Matsubara frequencies. Then, the pseudogap opening can be observed directly in the imaginary part of the self-energy. In the pseudogap regime, ImΣ bends towards negative large values approaching the zero (Matsubara) frequency limit from above, while in the Fermi-liquid regime the bending is always towards small values. The onset of a pseudogap can hence be detected by the slope of the imaginary parts of the self-energy between the first and second Matsubara frequency (see also Refs. [45,48]). We added this point to the revised version.

10) Concerning the problems associated to the form-factor truncation that affects the SBE formulation of the SDE and not its parquet formulation, we added a short paragraph at the end of Sec. 3 to clarify this point.

11) The explicit form of Eq. (84) in the revised version is now available in Ref. [57] for the extended Hubbard model, where the splitting between the bosonic and fermionic parts of the interaction is reported and discussed in detail.

Furthermore, we corrected some typos and modified some formulations to improve the presentation. We also shortened the discussion of the non-local case in App. B, since we realized that some parts would require a more detailed discussion. As it goes beyond the scope of the present work, we may rather address it in a forthcoming study.

We hope that the manuscript is now ready for publication.

---

## Round 1 · Referee Report · Anonymous (Referee 2) · 2025-1-17

Strengths

1 - clean derivation of the SDE in the newly introduced SBE formalism
2 - Application for a well studied case showcasing that it does work.
Since this reformulation of the SDE was not yet done and indeed offers a numerical advantage it definitely brings the application of the SBE decomposition to other cases closer to what is achievable.

Weaknesses

One of the main points the article makes, (which is even in the abstract) is
"In the application to the functional renormalization group, we find that the pseudogap opening in the two-dimensional Hubbard model at weak coupling is captured only in the magnetic channel representation of the SDE, while its expressions in terms of the density and superconducting channels fail to correctly account for the driving antiferromagnetic fluctuations."
I understand what the intention behind saying this is, but to a non expert reader not familiar with the matter, this reads like one should never do TU since the becomes choice dependent. I would advice the authors to stress that this is for their specific choice of form factors, which in itself is a non converged choice. The apparent ambiguity essentially means that the calculation is not converged. The authors also mention this at one point but keep underlining that this is a central issue.In a sense, this is not a bug but a feature due to the numerical setup.

Report

1) In your abstract you mention "At weak coupling, this effective bosonic interaction yields quantitatively accurate results, while the multiboson contributions are irrelevant and can be neglected" this implicitly assumes local interactions, since recently a work from the same group showed that for long range interactions the rest function is often non negligible.

2) Please rewrite the misleading statements I was referring to (I found these in the abstract, introduction and later at the discussion of the results)

3) Please introduce the objects the first time you introduce them, e.g. what do numbers as indices stand for (eq. 1), what is G, what is V (eq. 2) what is the difference between a | and a , between two indices?

4) Minor suggestion: Ref. 31 is behind a paywall and since you are already publishing in an open access journal it would be nice to have a second Reference here which is openly accessible deriving the equation.

5) Eq. 14, o I understand correctly that this equality essentially comes about due to all U-irreducible diagrams contained on the right hand side becoming by definition U reducible due to the tensor contraction? Or is there another intuitive understanding of this?

6) Again a general remark: I understand that the extremely compressed notation you use is helpful since it allows to write equations much quicker. However, for scientists or students new to the field it would be much easier to follow the derivation if there wasn't a new short hand for a specific product or a collapse of indices introduced on every second page early on. I do not want you to rewrite all of the paper, but I do want you to consider this for future publications. Another upside would be that you do not need to translate the final equations back.

7) Technically, the first introduction of a truncated unity as you use it was within the extended coupled ladder approach introduced in 1D real space systems, see https://arxiv.org/abs/1311.3210 and https://arxiv.org/abs/1609.07423, you might want to add citations to these.

8) The statement "In fact, the restriction to a small number of form factors, i.e., truncating the form factors before full convergence, leads to a violation of the crossing symmetry and is recovered only for a converged set, where the crossing symmetry is again fulfilled" copied from Ref. 42 is not true. Once you look into the Appendix there you find that the condition for the crossing to be fulfilled is essentially that $\sum_{l_3} f^{}{l_3,k_3'}f f^{}= \sum_{l_3}{l_3,q-k_3'}f$ are equal. Now assuming you use a plain wave form-factor basis, you can directly confirm that this holds true whenever for each bond $l_3$ the negative counterpart exists as well. Thus crossing symmetries are fulfilled as long as complete form factor shells are considered in the calculation.

9) Footnote on page 11 - does up to include the order or not? Please clarify.

10) Please explain how you are performing the numerical simulations - e.g. I assume that at each step of the flow you iterate the SDE till convergence but I did not find this anywhere.

Requested changes

see above

Recommendation

Ask for minor revision

  • validity: good
  • significance: good
  • originality: good
  • clarity: ok
  • formatting: good
  • grammar: good

Author:  Miriam Patricolo  on 2025-02-04  [id 5187]

(in reply to Report 2 on 2025-01-17)

We thank the Referee for his assessment and valuable suggestions that helped to improve the manuscript.

In the following, we address the points raised by the Referee:

1) The statement on the weak-coupling regime in the introduction refers indeed to local interactions. This applies in fact to the whole introduction, since in the literature only this case is treated. We here also refer exclusively to those works. When referring to the extension to non-local interactions we refer to our recent preprint. In that case the multiboson contribution can be indeed sizable, but its flow is still negligible in the regime considered in our analysis.

2) We reformulated the misleading statements on the different convergence in the number of form factors for the three channel representations. This is indeed an important point and we thank the referee for this comment.

3) Besides the citation of Ref. [9], we added further explanations of our notation below Eqs. (1) and (2). In our revised version, we avoid a comma in the indices (i.e., G_{0,1′|1} is replaced by G_{0;1′|1}) to emphasize that 0 or ph are not index labels, but refer to the type of the objects. The vertical bar | is used in accordance to Ref. [9] to separate indices belonging to creation and annihilation operators. Omitting it would not be harmful.

4) Concerning Ref. 31, we added a second reference where possible.

5) Equation (14) can be understood by separating the U -reducible vertex ∇r = ¯λr • wr • λr into U -reducible and U -irreducible parts (cf. Eqs. (34)–(36) in Ref. [9]). As you pointed out correctly, the product wr • λr is U reducible itself, but contains all U -irreducible contributions on the right, which can be linked to a U -reducible part on the left.

6) We used and extended the notation introduced in Ref. [9] to make the expressions more compact. In fact, the Schwinger–Dyson equation for the self-energy includes sums over Matsubara frequencies, momenta, and channel indices. This is made explicit in Eqs. (37), which are much longer and more complex than the compact forms Eqs. (20)–(21). For our derivations, it was essential to use the compact notation in order to keep the overview. We were careful about the definition of all the products (◦, •, ·). In the appendices A, C and D we provide the details to translate all equations introduced in the paper, in particular also the flow equations for the SBE objects and the self-energy, from our compact notation to their more explicit versions like Eqs. (36) or (37).

7) The idea of the coupled ladder approach in real space is certainly very similar, thank you for pointing this out to us. We included the references.

8) We thank the referee for pointing out the formula related to the fulfillment of the crossing symmetries. However, it seems to depend on the conventions used for the form factors. While we were able to verify that it holds for the non-symmetrized form factors, we could not show it to hold for the symmetrized ones used by us. Therefore, we will only make a more general statement on the violation of the crossing symmetry when a reduced set of form factors is used.

9) In the 2ℓ truncation, the 3rd order in U is accounted for correctly, corrections appear in U 4. We clarified this point in the footnote on p. 12.

10) We added a short paragraph specifying the details of the numerical simulations.

Furthermore, we corrected some typos and modified some formulations to improve the presentation. We also shortened the discussion of the non-local case in App. B, since we realized that some parts would require a more detailed discussion. As it goes beyond the scope of the present work, we may rather address it in a forthcoming study.

We hope that the manuscript is now ready for publication.

Anonymous on 2025-02-10  [id 5202]

(in reply to Miriam Patricolo on 2025-02-04 [id 5187])
Category:
remark

I thank the authors for taking the comments serious and I do think that the paper is now more accessible. One last remark, the symmetrized form-factors are a unitary transformation of the plain-wave form-factors, thus the proof for 8) goes along the same lines and boils down to again using the complete shell, otherwise the unitarity is not fulfilled. Thus the breaking of crossing symmetries is bound to the specific choice of only using an on-site and the $d_{x^2-y^2}$ form-factor.

---

## Editorial Decision

published